# Intracellular Membrane Transport in Vascular Endothelial Cells

**DOI:** 10.3390/ijms24065791

**Published:** 2023-03-17

**Authors:** Alexander A. Mironov, Anna Mironov, Barbara Sanavio, Silke Krol, Galina V. Beznoussenko

**Affiliations:** 1The AIRC Institute of Molecular Oncology, 20139 Milan, Italy; 2Sacco Hospital, Universita degli Studi di Milano, 20122 Milan, Italy; 3Nanomedicine Laboratory, Fondazione IRCCS Istituto Neurologico Carlo Besta, c/o AMADEOLAB, 20133 Milan, Italy; 4Encytos BV, 7511 JE Enschede, The Netherlands; 5MESA+ Laboratory, University of Twente, 7522 NH Enschede, The Netherlands

**Keywords:** intracellular transport, endothelial cells, transcytosis across endothelial cells, the Golgi complex, micropinocytic vesicles, intra-Golgi transport, the kiss-and-run model, vesicular–vacuolar organelle

## Abstract

The main component of blood and lymphatic vessels is the endothelium covering their luminal surface. It plays a significant role in many cardiovascular diseases. Tremendous progress has been made in deciphering of molecular mechanisms involved into intracellular transport. However, molecular machines are mostly characterized in vitro. It is important to adapt this knowledge to the situation existing in tissues and organs. Moreover, contradictions have accumulated within the field related to the function of endothelial cells (ECs) and their trans-endothelial pathways. This has induced necessity for the re-evaluation of several mechanisms related to the function of vascular ECs and intracellular transport and transcytosis there. Here, we analyze available data related to intracellular transport within ECs and re-examine several hypotheses about the role of different mechanisms in transcytosis across ECs. We propose a new classification of vascular endothelium and hypotheses related to the functional role of caveolae and mechanisms of lipid transport through ECs.

## 1. Introduction

Endothelial cells (ECs) line the luminal surface of all blood and lymphatic vessels and sinuses of lymphatic nodes [1]. ECs form a monolayer and function as a semipermeable barrier, which regulates the flux of fluid, proteins, and blood cells across the vascular wall into parenchymal tissue and maintains the antithrombotic and anti-inflammatory state of the EC surface [2]. ECs are one of the most common cells in the human body. Their aggregated volume in blood and lymphatic vessels is equal to that of the liver.

The role of the endothelium in humans is enormous. Significant progress has been made in the deciphering of the molecular mechanisms responsible for the endothelium’s function and its involvement in diseases. However, most of the knowledge on these mechanisms was obtained from cell culture and there is a necessity to re-examine whether all these mechanisms are similarly involved in the function of the endothelium in adult organisms and tissues. Fast development of light and electron microscopy has allowed us to examine EC directly in tissues or to combine different microscopical methods [3,4]. In most papers on ECs the main topic for analysis and discussion is molecular mechanisms. Here, we try to re-examine information about the functional morphology of ECs in tissues. Non-vascular ECs are not discussed here. Additionally, molecular mechanisms involved in the organization and function of ECs will only be briefly described.

## 2. Architecture of Endothelium

ECs have several morphological features that are common to all locations: they are polarized and very flat (with the exception of a simple cuboidal epithelium in lymph nodes), and serve as a barrier between the blood and the interstitial space. In vitro, typical endothelial cells are small, round to polygonal shaped, and are arranged uniformly. Their diameter ranges from 50 to 70 µm [5]. Arterial ECs are narrow and elongated in the direction of flow, whereas venous endothelial cells are generally shorter and wider [6]. In arteries, where blood flows without turbulence, ECs have an ellipsoidal shape, and are aligned coaxially [4]. As the diameter of the vessel decreases, the flat shape of the EC increasingly approaches the shape of a flattened plastic covering of a cylinder of different diameters. Each EC exhibits a clear division into central and peripheral zones (Figure 1A–F). In the center there are a small number of mitochondria, the Golgi complex (GC), endosomes, and the endoplasmic reticulum (ER; Figure 1G–I).

The EC peripheral zone is very thin and, in some places, does not exceed 200 nm. Usually, in the peripheral zones of multicaveolar continuous ECs (see classification of ECs below), for example, in the ECs of the blood capillaries of the lungs, there are a lot of caveolae, and in the fenestrated ECs of the intestinal villi, endocrine organs, and peritubular blood capillaries of the kidneys there are many fenestrae (Figure 2A,C–E). However, in high endothelial venules, ECs do not contain typical peripheral zones (Figure 1A,B). Fields of the peripheral zones with increased numerical density of fenestrae are separated by strands of thicker cytoplasm (Figure 2A). The peripheral zone of the EC of lymphatic vessels is also thin and contains a small number of caveolae, but does not contain fenestra, pores, or caveolae (Figure 1H,I). In the sinusoids of the spleen, the intercellular contacts are partially open (Figure 1C).

In arteries, there are always a number of outgrowths (microvilli or lamellipodia) along the contacts between ECs (Figure 1D). These outgrowths serve as a membrane reserve during the passage of a blood bolus which induces widening of the arterial lumen. Since there are no pulsations of intravascular pressure in the veins, the severity of cytoplasmic outgrowths located along the inter-endothelial contacts is significantly lower (Figure 1F) (see also Figure 6A presented by Barboriak et al. [7]; Figure 3A presented by Musalam et al. [8]; Figure 3: Native SV presented by Liu et al. [9]; Figures 144 and 145 presented by Volkova et al. [10]). However, similar outgrowths are observed in ECs in the blood capillaries of the brain (Figure 3H).

In arterial ECs, the microtubule organizing center is localized downstream of the nucleus. Large veins exhibit the opposite localization, namely, upstream of the nucleus [11,12,13]. Similar polarization patterns have been reported in the localization of the perinuclear Golgi apparatus, which is localized upstream of bovine endothelial cells grown under laminar shear stress in culture [14]. Axial polarization along a vessel of the GC upstream of nuclei has also been observed in ECs of live zebrafish embryos using a fluorescent reporter. The GC is localized upstream of endothelial nuclei in the dorsal aorta and branching arteries [15]. In contrast, in the posterior cardinal vein, where shear stress is much lower, ECs do not exhibit clear polarization of the GC [13,15].

In lymphatic vessels, ECs exhibit two phenotypes. The first type of lymphatic EC is LYVE-1-positive and their contacts are sinusoid-like. They line lymphatic capillaries and the visceral ECs of lymph nodes. The do not contain intermittent tight junctions (TJs), and exhibit standard AJs situated near TJs and the point AJs on the tip of their lamellipodia covered the neighboring ECs. The second type of lymphatic EC is LYVE-1-negative. These ECs line lymphatic post-capillaries, vessels containing intraluminal valves and parietal ECs in lymph nodes. Near-contact outgrowths and microvilli in the EC of the thoracic lymphatic duct are needed in order to increase the surface area of the EC when the lymphangions of this vessel contract [10] (Figure 1E).

**Figure 1 ijms-24-05791-f001:**
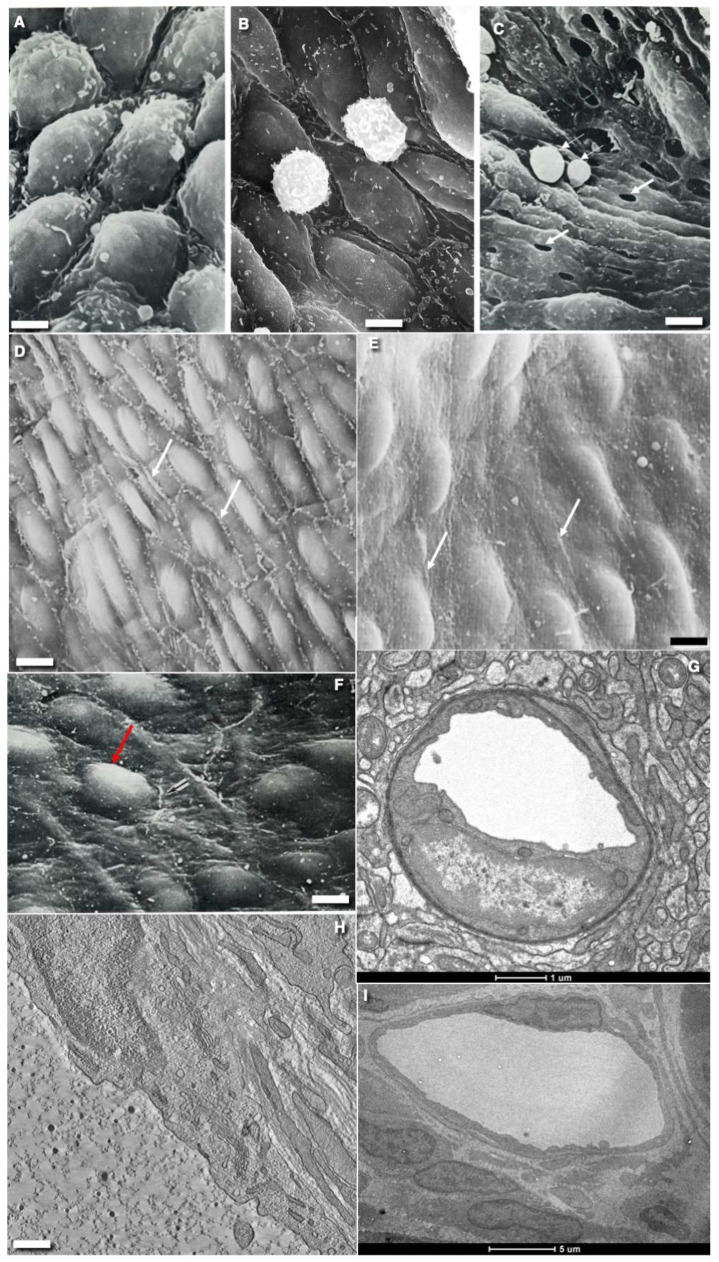
Structure of different types of endothelial cell from rats and mice. (**A**,**B**) Scanning electron microscopic images of the luminal surface of ECs from the high endothelial venule in lymphatic node. (**C**) SEM of the luminal surface of the endothelium with open contact in rat spleen. White arrows show erythrocytes. (**D**) Scanning electron microscopic image of the luminal surface of rat aorta after perfusion fixation. Cytoplasmic outgrowth (white arrows) are visible along the EC contacts. (**E**) Scanning electron microscopic image of the surface of EGs of the lymphatic thoracic duct. White arrows show near-contact protrusions. (**F**) Scanning electron microscopic image of the luminal surface of vena cava inferior. Black arrow shows rare outgrowths along intercellular contacts. Protrusions/elevation of nuclei of EC are shown with red arrows. Outgrowths along cell contacts are almost invisible. (**G**) Cross section of the brain blood capillary. ECs contain low number of caveolae. (**H**) The virtual EM tomo-slice of cross section of central zone of EC from lymphatic capillary of small intestine villus after starvation. The PM of ECs have only a few caveolae. (**I**) The ECs of lymphatic vessels are divided into nuclear and peripheral zones. Images (**A**,**B**,**F**) are from Figures 142, 170 and 144 (correspondingly) by Volkova et al. [10]. Images (**D**,**E**,**H**) are taken from Figure 2A and supplementary images (correspondingly) presented by Sesorova et al. [16]. The image (**G**) is taken from Figure 3E presented by Cottarelli et al. [17]. The image (**I**) is taken form Figure 5I by Sesorova et al. [18] in agreement with the CC By license (the Creative Commons Attribution 4.0 International License). Scale bars: 12.2 µm (**A**,**B**); 18.5 µm (**C**); 11.2 µm (**D**); 6.7 µm (**E**); 13.1 µm (**F**); 1 µm (**G**); 650 nm (**H**); 5 µm (**I**).

**Figure 2 ijms-24-05791-f002:**
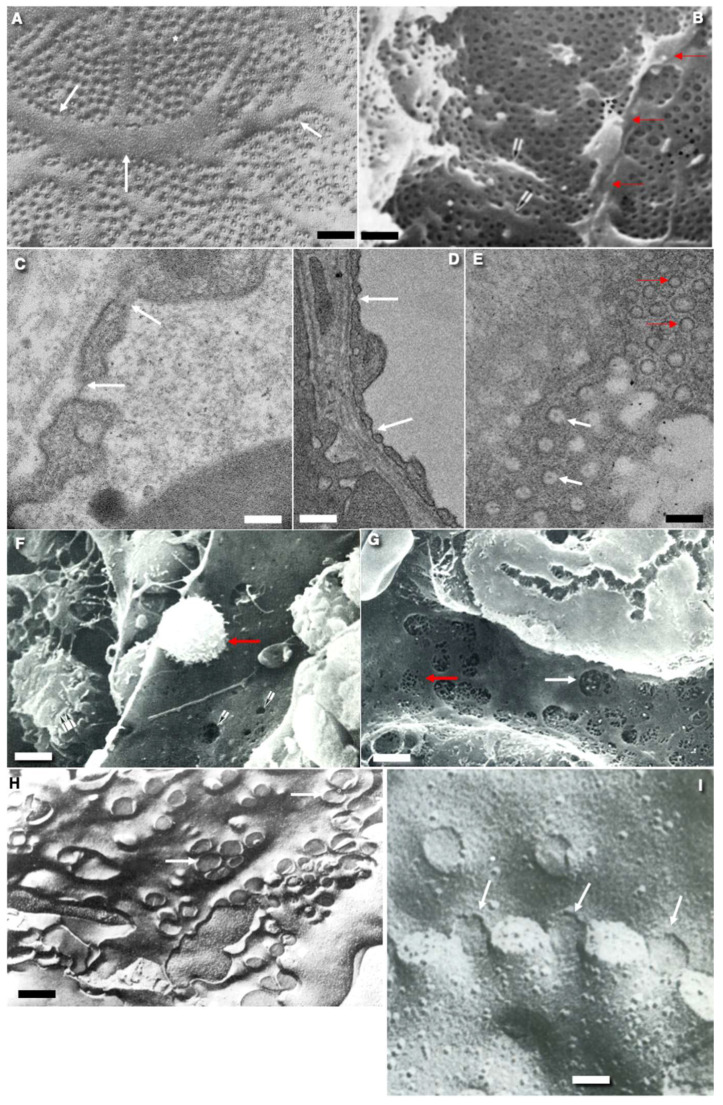
Structure of transcytosis compartments of rat endothelial cells. (**A**) Tangential freeze-fracture of the endothelium of blood capillary in frozen intestinal villus. White asterisks indicate the fields of fenestrae. White arrows demonstrate the thick zone of endothelial cytoplasm between the fields of fenestrae. (**B**) Scanning electron microscopy (SEM) of the surface of the thin endothelium with small pores in the glomerular blood capillary. Red arrows indicate the intercellular contact. (**C**,**D**) Thin fenestrated endothelium. Transmission electron microscopic (TEM) sections across the blood capillary of intestinal villus. White arrows demonstrate fenestrae. Red arrows show cross section of caveolae. (**E**) The tangential TEM section of the thin fenestrated EC. White arrows indicate fenestrae. Red arrows show cross sections of caveolae. (**F**) Scanning electron microscopic image of the lumina surface of the sinusoid in rat red bone marrow. White arrows indicate large pores. Red arrow shows lymphocyte. (**G**) SEM of the luminal surface of blood sinusoid in liver. White arrow shows large pore. Red arrow demonstrates the field of small pores. (**H**) Tangential fracture of frozen liver blood sinusoid. White arrows shows large pores. (**I**) Tangential cross-fracture (after quick freezing) of the fenestrated ECs. White arrows show localization of fenestrae. Images (**A**,**H**,**I**) are taken from Figures 22c/в), 21e/д), and 23b (correspondingly) presented by Komissarchik and Mironov [19]. Images (**B**,**F**,**G**) are taken from Figures 125, 152 and 126 (correspondingly) presented by Volkova et al. [10]. The images (**C**–**E**) are taken from Figure 6D and Supplementary images presented by Sesorova et al. [18] in agreement with the CC By license. Scale bars: 1.7 µm (**A**,**B**); 100 nm (**C**); 170 nm (**D**); 130 nm (**E**); 5 µm (**F**); 7 µm (**G**); 1.7 m (**H**); 80 nm (**I**).

**Figure 3 ijms-24-05791-f003:**
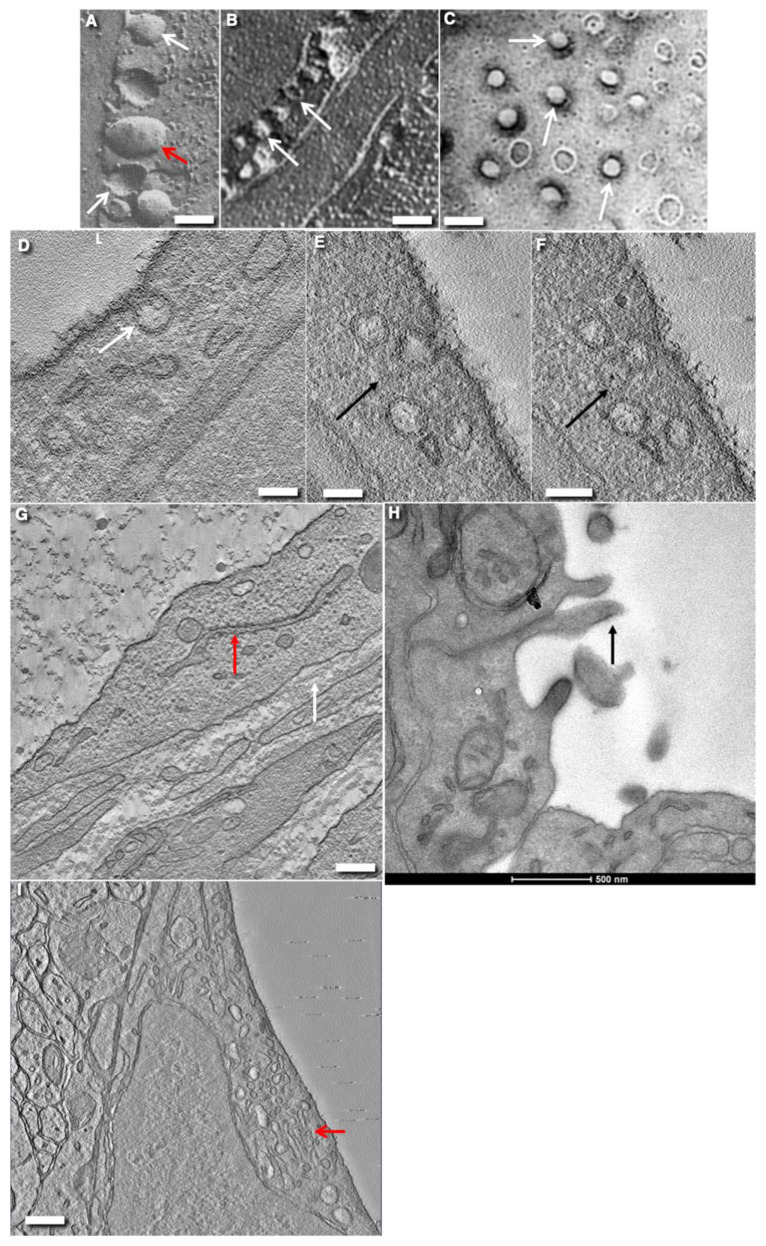
Structure of endothelial cells in different types vessels in rats and mice. (**A**,**B**) The cross-freeze-fractures of ECs in the capillary of the heart. Platinum replicas obtained after freeze-fracture and evaporation of platinum of closed multicaveolar endothelium. White arrows show fractures of caveolae on the perpendicular fracture of the capillary. Red arrow in (**B**) demonstrates the presumable fracture of clathrin-coated bud. (**C**) Tangential fracture of the frozen thin multi-caveolar endothelium with closed contact from the heart blood capillary. Evaporation with platinum during rotation of the sample. White arrows indicate fracture of caveolae. (**D**) White arrow shows the rare caveolae in cross section of EC in brain blood capillary. (**D**–**F**) Serial virtual EM tomo–slices of the caveolae (**D**, white arrow) and caveosome (**E**,**F**, black arrows) which appears like a bunch of grapes. (**G**) The tomography section of EC of lymphatic thoracic duct. No caveolae are visible. Basement membrane is shown with the white arrow. The ER is shown with the red arrow. (**H**) Cross section of EC from brain blood capillary. Caveolae are not visible. Arrow shows near-contact protrusions. (**I**) Mouse brain. Virtual EM tomo-section of the central zone of EC from brain blood capillary. The GC is shown with red arrow. Caveolae are not visible. Images (**A**–**C**) are taken from Figures 16a, 17c/в, and 20d/r (correspondingly) presented by Komissarchik and Mironov [19]. The image (**H**) is taken from Figure S3 used for Supplementary Materials but not included in the text from Sesorova et al. [18]. Images (**H** and **I**) are taken from Supplementary Materials presented by Cottarelli et al. [17] but not included into the final text in agreement with the CC By license. Scale bars: 60 nm (**A**); 120 nm (**B**); 110 nm (**C**) 55 nm (**D**); 70 nm (**E**,**F**); 450 nm (**G**); 500 nm (**H**); 290 nm (**I**).

## 3. Compartments of Intracellular Transport in ECs

In general, ECs in cell culture are similar to other types of polarized cells. However, the level of their polarization is much lower than, for example, that of MDCK cells. Similarly, no one has yet described any features in the molecular organization of EC transport routes. A review describing the intracellular transport during angiogenesis was published by Francis and Kushner [20]. Description and images of the GC in ECs are presented by Neumuller and Ellinger [21].

In ECs, the general organization of the secretory pathway seems to be similar to that of other flat polarized cells (Figure 1H and Figure 4A–D). Membrane and secretory proteins are synthesized in the ER. Mechanisms of ER–Golgi transport in ECs have not yet been examined. However, it seems that these mechanisms are similar to those observed in other cell types, especially if one takes into consideration new models of ER–Golgi transport [22,23,24,25,26].

COPII buds are not found in quiescent ECs of adult organisms [16]. They are observed only in regenerating ECs when the EC is in the G2 phase of the cell cycle. During this phase, ECs synthesize membrane proteins for their consecutive division [4,16,27]. Similarly, in other tissue cells of adult organisms, COPII buds are found only in cells during their G2 phase or in cells forming large secretory granules, namely, in fibroblasts, in pancreas, and in goblet cells [28,29,30,31]. For instance, in the liver, there are no secretory granules and therefore very low density lipoproteins (VLDL) are transported towards the Disses space in Golgi-to-PM carriers. In enterocytes of the whole small intestine, ER exit sites are not found [4,16,27,32]. After inhibition of COPII, chylomicrons are still formed in the smooth ER [32]. In enterocytes, COPII-coated buds are not observed after starvation [4], only after the birth and even after initiation of lipid transcytosis by the delivery of lipids into the intestinal lumen [27].

ECs contain the typical GC composed of 3-4 medial cisternae (Figure 1H, Figure 3I and Figure 4A,B). The shape of the GC is similar to the resting GCs, which are described in other cell types [4,31]. Most Golgi stacks are not covered by the cis-most cisterna (CMC) and the trans-most cisterna (TMC). However, it is not clear whether the GC of ECs forms a ribbon [33] or whether it is composed of peripheral [34] or central fragments [23,35]. ECs secrete components of BM in the Golgi-to-BLPM carriers [16]. Of interest, the externally added lectin WGA cannot reach the GC of ECs [21]. This observation also suggests that the GC is in a resting state.

ECs have unique secretory granules called Weibel–Palade bodies (WPB) with a length of up to 5 µm and a diameter ranging from 100 to 300 nm [36]. WPBs are connected with the TGN. WPBs are cigar-shaped secretory granules. The most predominant protein found in this structure is pro-thrombotic von Willebrand factor (VWF). This large multimeric protein is capable of initiating the clotting cascade. WPBs are formed at the acidic trans-Golgi network and are secreted through the apical plasma membrane (APM) [37].

The formation of WPB requires adaptor protein 1 and clathrin [38,39]. Then, the clathrin coat disappears from the surface of matured multivesicular bodies [21]. Some CD63/lamp3 is also present in WPBs [40]. VWF and WPBs are used as markers of ECs in culture and in situ [20]. The tetraspanin CD63/lamp3 distributes predominantly to the internal membranes of late endosomes, which contain the unique lipid lyso-bis-phosphatidic acid, and cycles between endocytic and secretory compartments. Mechanisms responsible for the Golgi-to-APM transport are unknown. It seems that the content of WPBs is secreted according to the regulated secretion mechanisms [28]. VSVG, asialoglycoprotein receptor, and tumor necrosis factor are delivered towards the BLPM. ECs secrete proteins that form their basement membranes (BM), namely, fibronectin, procollagen III, procollagen IV, collagen VII, heparan sulphate proteoglycan, laminin, integrins, entactins, fibrilin, and dystroglycans, through the BLPM [21,41,42].

There are several markers of ECs. VWF and WPBs are used as markers of ECs in culture and in situ. CD157 is expressed in endothelial cells in large arteries and veins, but not in capillaries of other mouse organs. In the liver, CD157 is almost exclusively expressed with CD200 [43,44]. D2-40, LYVE-1, Prox-1, VEGF-3 are routinely used markers of ECs in lymphatic capillaries and lymphatic vessels [45].

## 4. Endocytosis in ECs

As in other cell types in ECs, endocytosis is involved in the internalization of molecules from the plasma membrane and extracellular environment, and plasma membrane recycling, including uptake and the degradation of signal molecules [46,47]. The development of endosomes depends on the role they play in transcytosis through the central zone of EC. (Figure 4A,B).

The main components of the endocytic system are clathrin-dependent endocytic buds (“vesicles”), and early and late endosomes and their derivatives [46]. Additionally, ECs exhibit a clathrin-independent endocytosis with its own less characterized components. The clathrin-dependent pathway is responsible for transferrin transport through brain ECs [47]. Its receptor is on the APM. It determines direction of transcytosis [48]. Clathrin-coated buds are observed mostly within the central zone of EC. These kinds of buds with well-developed necks are more often observed than clathrin-coated dome-shaped invaginations on the PM. According to the current consensus, ECs could use both the clathrin-dependent and independent pathways. The latter are classified as (1) caveolae-mediated endocytosis, (2) clathrin- and caveolae-independent endocytosis, and (3) micropinocytosis [49] (Figure 4A,B).

After budding, fission and uncoating clathrin-dependent endocytic vesicles fuse with each other or with pre-existing early endosomes. The first event also results in the formation of early endosomes, which within the next 2–3 min undergo segregation into vacuolar head and tubules connected with it, where the main mass of lipid and proteins destined for recycling to PM is accumulated, although soluble contents are concentrated in vesicular regions due to their greater fractions volume. In the GC of ECs, multivesicular bodies (MVBs) are small [47]. Similarly, the mechanisms responsible for the transport along the endocytic pathways in ECs, namely, vesicular, maturation, or kiss-an-run mechanisms, are mostly unknown.

## 5. Endothelial Polarity

ECs are less polarized than other monolayered epithelia [50]. The PM of ECs is divided into two main parts, namely: APM (or luminal PM), which faces the aqueous phase (i.e., blood), and basolateral PM (BLPM; or abluminal PM), which faces and is attached to structures of the extracellular matrix. In contrast to other polarized epithelial cells, where the APM area is usually smaller than the BLPM area, in ECs of blood and lymphatic capillaries, the size of the APM and BLPM is almost equal [41].

The APM of ECs is covered by a layer of polysaccharides, the glycocalyx (Figure 5F), composed of glycoproteins and proteoglycans such as syndecan-1, -2, -4 and glypican [51]. The development of the glycocalyx differs in different vessels. The glycocalyx thickness varies from 20 to 50 nm [41,52]. Endothelial glycocalyx decreases water, albumin, and LDL transcytosis. Chronic administration of hyaluronidase leads to degradation of glycocalyx, and albumin adheres to podocytes [53].

In culture, under specific conditions after quick freezing-cryo-substitution, the thickness of glycocalyx of ECs can be up to 11 µm [54]. The endothelial glycocalyx is the polysaccharide gel that covers the luminal surface of the endothelium and acts as a filtration barrier at the fenestrated endothelium in the glomeruli [55,56]. Glycocalyx exhibits a net negative charge and consists of glycosaminoglycans (heparan sulfate, chondroitin sulfate, and hyaluronan) and associated proteins, glycolipids, and glycoproteins. Due to the presence of sialomucins, glycoproteins, CD34, and PODXL containing a lot of negatively charged glycosyl residues, the luminal surface of ECs is anti-thrombogenic [57]. ECs could be easily transformed into non-polarized cells [50].

The long carbohydrate polymer hyaluronan accumulates water and acquires the gel-like properties of this layer [58,59,60,61]. For instance, pores in the glomerular endothelium are not empty, but are filled with densely packed hyaluronan that anchors the glycocalyx to the glomerular basement membrane [55,58]. This glycocalyx acts as an almost perfect barrier against albumin filtration [58,62,63]. However, large, straight nanotubes (200–300 nm) can be partially filtered [58,64]. The luminal (apical) surface of human aortic ECs contain rare cilia [65].

There are several markers of the APM of ECs, namely, CD34, ERM protein, sialomucins consisting of CD34, and PODXL/PODXL2 [57,66,67,68,69]. Markers of APM and BLPM in the ECs of brain blood capillaries are described by Worzfeld and Schwaninger [48]. However, the molecular basis of apicobasal polarity in quiescent ECs is not well understood. Vascular endothelial growth factor, histamine, and insulin-like growth factor-binding protein 3 increase the trans-endothelial permeability of the blood–brain barrier when administered to the BLPM. The polarized response to vascular endothelial growth factor is found exclusively in the brain and retina, but not in peripheral endothelial cells, highlighting the special status of endothelial cells in the central nervous system. ECs secrete von Willebrand factor. RAC1 is enriched baso-laterally. Crumbs proteins are apically localized [48].

Additionally, there are several other proteins localized within APM, namely, interleukin-8, the tissue type plasminogen activator, the endothelin-1 and the endothelin-converting enzyme, eotaxin-3, angiopoietin-2, and osteoprotegerin, as well as integral membrane proteins such as the adhesion molecule P-selectin, the tetraspanin CD63, and the fucosyl transferase VI as well as Rab27A and Rab3D [21,69]. The partition-defective (Par) polarity complex (Par3/Par6/atypical protein kinase C (aPKC]) has also been implicated in apical polarity in epithelial cells and ECs, and disruption of the Par complex is strongly associated with failed lumen formation [69,70,71]. In the ECs of capillaries, paxillin immunostaining was found to be higher than in other types of ECs [72].

ECs contain receptors specific for LDL, ß-VLDL, transferrin, insulin, albumin, ceruloplasmin, transcobalamin, and other plasma molecules on the apical and basal membrane of ECs [41,73]. Scavenger receptor class B, type I (SR-BI) is found in APM caveolae of brain capillary endothelial cells. Disruption of caveolae with methyl-beta-cyclodextrin resulted in the mis-sorting of SR-BI to the basolateral membrane [74]. In PODXL-null mice, moesin no longer localizes to the apical membrane [75]. CD34 sialomucins make the APM of ECs anti-thrombogenic due to their negatively charged residues, because its ectodomain is highly sialylated [57,69,76]. How these markers are transported along the secretory pathway of ECs after their synthesis in the ER is not clear [77,78].

Endothelial near-contact microvilli (Figure 1D,E) were first described by Gabbiani and Majno [79]) as dynamic structures whose surface density and length are affected by pulse flow. However, a complete elucidation about the role of these microvilli is still lacking, although their involvement in blood flow dynamics, leukocyte recruitment and bacterial internalization has been reported.

**Figure 5 ijms-24-05791-f005:**
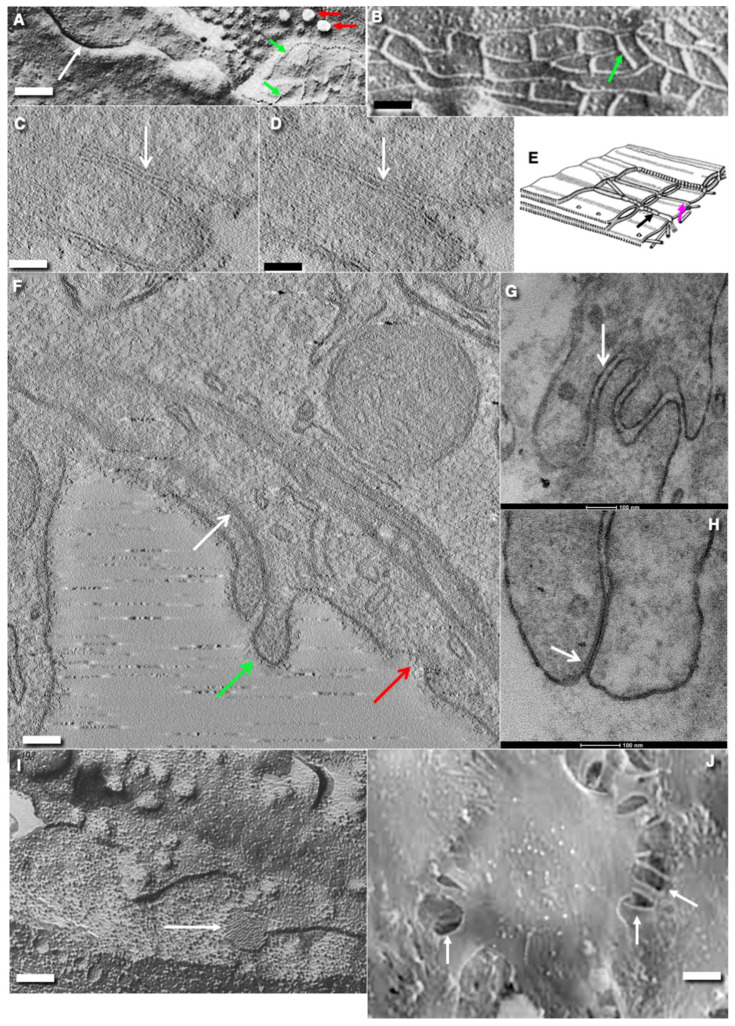
Structure of the EC contacts and junctions in rats and mice. (**A**) ECs of blood capillary in derma. Platinum replicas obtained after tangential freeze-fracture of the EC near the inter-endothelial contact. Green arrow indicates the fracture of tight junction. Red arrow demonstrates the fracture of caveolae. (**B**) The freeze-fracture of the typical TJ between ECs of blood brain capillary. Green arrows indicate the locking fibril of TJ. (**C**,**D**) Serial virtual tomography electron microscopic slices. White arrows show the locking fibrils of TJ which appears as a white dot connecting two membranes of opposite ECs. (**E**) Scheme of TJ. On the freeze-fracture, of the contact zone of ECs, locking fibrils consisting of intramembrane particles appear as widely anastomosing and fence-like elevations (ridges; black arrow)) or depressions (grooves; magenta arrow). (**F**) The tile-like contact between ECs. It is supplemented with TJ (white arrow). The green arrow indicates the near-contact cytoplasmic protrusions between ECs of blood capillary in brain. Red arrow shows clathrin-coated pit on APM of EC. On the surface of APM the glycocalyx is visible. (**G**) Interdigitating contact between ECs of lymphatic thoracic duct. (**H**) Gap junctions (arrow) between ECs of rat thoracis lymphatic duct on the routine transmission EM section. (**I**) Freeze fracture of gap junction (white arrow) between two ECs of rat aorta. Gap junctions appear as an aggregate of intramembrane particles. (**J**) Open contacts (arrows) between ECs of rat aorta within the zone of the turbulent blood flow. Images (**A**,**B**,**E**) are taken from Figure 25 presented by Komissarchik and Mironov [19]. Images (**C**,**D**) are taken from Figure 5F by Kakogiannos et al. [80]. The image in agreement with the CC By license. The mage (**J**) is taken from Figure 3G by Mironov et al. [6] in agreement with the CC By license. Scale bars: 165 nm (**A**); 70 nm (**B**); 65 nm (**C**,**D**); 80 nm (**F**); 110 nm; 80 nm (**H**); 100 nm (**I**); 5.1 µm (**J**).

## 6. Basement Membrane of ECs

The abluminal PM or BLPM of ECs is attached to the BM (Figure 4D,E). However, ECs of sinusoids in the liver and ECs in lymphatic capillaries have no evident continuous BM [5,18,81,82] (see below). The BM of continuous, fenestrated, and low-size porous endothelium is well developed. In the aorta, due to the fact that the wall and BM are constantly distended and then relaxed, BM is mesh-like rather than solid [6]. The BM proteins are secreted through BLPM, whereas proteins providing anti-thrombogenicity of ECs are secreted through APM [16,83,84,85]. In particular, integrins are delivered to the BLPM, and highly glycosylated proteins forming the glycocalyx are delivered to the APM.

The mesh-like (see below) BM becomes multilamellar (Figure 4E; see [16]) in areas of turbulent flow, in old and hypertensive animals, and after repetitive re-endothelization [16,65,86] (Figure 4D,E). The protein composition of BMs changes with age and when ECs are damaged and then regenerate [87]. Additionally, an age-dependent increase in BM thickness has also been reported for the human vascular BMs by other groups [87,88,89]. However, in places of increased hemodynamic stress in the rat aorta, the mitotic index is not increased [4,90]. This predisposes to development of atherosclerosis [6].

In some lymphatic capillaries, BM is absent only in the zones where there are no TJs between ECs [81,91]. Additionally, BM is absent in blood vessels during early stages of embryogenesis (i.e., in 16- to 18-somite embryos, the endothelium lacked a basal lamina [92]), in sprouts during initial stages of angiogenesis, and in angiosarcomas where no BM around blood vessels is observed [93]. The endothelium in developing bones lacks the typical BM [94]. In all of these cases, the polarity of ECs is limited because the EC polarization of ECs is not possible without attachment of ECs to the substrate by adhesion sites [95]. The thickness of the EC BM varies from 30 nm to 200 nm [96].

The endothelial side of BMs is characterized by proteins that promote epithelial cell adhesion of ECs and their migration. The stromal side of BMs is endowed with proteins or carbohydrate epitopes that are anti-adhesive, but bind stromal-typical ECM proteins and, thereby anchoring the BMs firmly with the adjacent connective tissues [87]. BM consists of collagen type IV, perlecan, the laminins 411 and 511, and other proteins. BM is also composed of integrins, heparan sulphate proteoglycans, and nidogens [97,98]. The BM of ECs also contains BM40 (osteonectin, SPARC), fibulin 1 (BM90) and 2, collagen types VIII, XV, and XVIII, and thrombospondin 1 and 2 [96]. Collagen IV binds to ß1 integrins (α1ß1, α2ß1). Activation of ß1 integrin by collagen results in RAC1 activation, which leads to the secretion of laminin and assembly of the basement membrane [48,94,99].

Two main components of BM, laminin and collagen IV, are large multidomain glycoproteins with a length of 120 nm and 400 nm, respectively [87]. All these proteins are synthesized in the ER of ECs and are then transported through the GC. Proteins of BM are secreted by ECs. During regeneration, ECs are synthesized components of BM being in the G2 phase [4].

In the ECs of lymphatic capillaries of small intestinal villi, BM is fragmented. Its isolated domains are visible near the inter-endothelial contacts containing TJs.

Pericytes are situated between two layers of BM. There is an important influence of pericytes on the basolateral/abluminal side of brain ECs [13,100,101]. ECs and pericytes form so-called peg-and-socket junctions containing N-cadherin [13,102]. Pericytes tightly regulate function and morphology of ECs, at least in the brain [100,102].

Secretion of the BM components occurred during the G2 phase. Multiple regenerations, as well as arterial hypertension and aging, also led to the thickening of the BM. Under these conditions, the speed of re-endothelialization increased. The thick BM captured more LDLs. LDLs formed after overloading of rats with lipids acquired higher affinity to the BM, presumably due to the prolonged transport of chylomicrons through neuraminidase-positive endo-lysosomes [16].

## 7. Inter-Endothelial Contacts

As with all other epithelial tissue, endothelium depends on its intercellular contacts. On cross sections, contacts between ECs can be interdigitating or tile-like (Figure 5F,G). In large arteries, the contacts in the areas of turbulent flow can be open (Figure 5J). On tangential freeze-fracture of EC, on contact in the blood capillary of the diaphragm the projection of contact line exhibits a sinusoidal shape [103]. On tangential projections of ECs in lymph capillaries, contacts have sinusoid-like shape [81,91,104]). EC contacts contain typical tight junctions (TJs; Figure 5A–E), and adhesive and gap junctions (Figure 5G,H). In ECs, gap junctions are rare and small (Figure 5H,I).

Tight junctions (TJs) are especially important for endothelial apical-basal polarity [77] (Figure 5A–E). TJs are composed of a branching network of sealing strands (crests). TJs prevent lipid diffusion along the external leaflet of PMs [13]. The efficiency of the TJ barrier increases exponentially with the number of strands [104]. The para-cellular transport pathway has been shown to be restricted to macromolecules in the range of 3–6 nm in size [105,106]. TJs between the ECs are found almost everywhere, but in the lymphatic capillaries they are fragmented and separated by zones where they do not exist.

In postcapillary venules where the exchange between blood and tissues is more dynamic, TJs are less developed than in arteries [41,107]. In the ECs of the spleen sinusoids, TJs are also divided into segments [10,108], between which there are open spaces used by erythrocytes for migration from the vessel lumen into red pulp and back. Importantly, only functional red blood cells able to deform their shape can penetrate back.

The morphology of the intercellular contacts in ECs differs from that of many epithelia, as TJs are not only located on the apical side, but may also be interdigitating (Figure 5G). In the brain, breaks in the TJ bands lead to oedema [80]). In adult mice, endothelial-specific deletion of Unc5B, a regulator of TJs, leads to leakage from brain capillaries [109]. During mitosis, contacts between aortic ECs become open [16,110,111].

In general, molecular organization of endothelial junctions is similar to that in other epithelial cells, with some modification [104]. For example, VE-cadherin, claudin-5, and PECAM-1 have been found in endothelial cells but not in epithelial cells. The inter-endothelial tight junctions are composed of transmembrane (claudins, occludin, and junctional adhesion molecule (JAM)-A) and cytoplasmic (zonula occludent (ZO)-1 and ZO-2, cingulin, AF-6, and 7H6) proteins linked to the actin cytoskeleton [80,112,113,114]. The endothelial cell-specific claudin-5 and claudin-3 have been shown to localize to endothelial tight junctions in the CNS of mice and man [115]. ZO-1 depletion led to tight junction disruption. Claudin-5 depletion only mimicked ZO-1 effects on barrier formation. ZO-1 controls endothelial adherent junctions, cell–cell tension, angiogenesis, and barrier formation [116]. When one of the proteins involved in the generation of tight junctions is affected, these junctions become less stable. For instance, Par complex (Par3-Par6-aPKC) promotes endothelial polarization; it associates with TJs and adhesive junctions (AJs) [95]. JAM-A acts via C/EBP-α to promote claudin-5 expression and improve the barrier function of ECs [80].

In ECs of capillaries, usually only one TJ strand is situated between APM and BLPM (Figure 5). Occludins are less abundant in the blood capillaries of adipose muscular tissues. In brain blood capillaries, the TJ fibrils are better developed and contain higher concentration of occludin and tricellulin [117]. It is known that occludin and tricellulin facilitate formation of an anastomosing tight-junction strand network [117]. TJs prevent this diffusion of lipids along the external leaflet of the PM.

Formation of adherent junctions precedes formation of TJs. AJs are usually located near TJs and contain vascular endothelial cadherin (VE-cadherin) molecules, which are connected with the actin cytoskeleton through associated a, ß, and p120 catenins [118,119]. Therefore, in VE-cadherin-depleted zebrafish, the distribution of ZO-1-containing junctions was disorganized and this induces impairment of the formation of lumen [95]. In VE-cadherin null mouse DAs, apical markers such as CD34, PODXL, and moesin fail to localize to the cell–cell contact, and consequently lumens do not form [69,75].

Numerous proteins, such as ß-catenin and N-cadherin, among others, constitute adhesive junctions. Domains with a high density of vascular endothelial cadherin (also known as Cdh5) and Rap1 were observed within contacts between ECs [69,78]. In the absence of VE-cadherin, apical and basal markers are disorganized. Depletion of Par3 and Par6 in HUVECs embedded in collagen matrices results in failed intracellular lumen formation, whereas chemical inhibition of PKC in the mouse prevents moesin phosphorylation and subsequent lumen generation [75,120]. Egfl7 is secreted into ECM and is required for initial establishment of lumens, but its specific function is still unknown. Casz1 binds and maintains Egfl7 expression in ECs when vessels remodel lumens during branching and fusion [69]. In ECs, the basolateral marker VE-cadherin localizes to adherent junctions, which establishes polarity [69]. In ECs, TJs become open during the EC mitosis [16]. However, the mitotic index of ECs in blood capillaries with continuous endothelium is very low.

PV1 (plasmalemmal vesicle-associated protein-1) is a 60-kD glycoprotein expressed in endothelial cells. Loss of PV1 specifically in ECs increases lung vascular permeability to fluid and protein. Endothelial-specific PV1 deletion also increased caveolae-mediated uptake of tracer albumin compared with controls, promoted Au-albumin accumulation in the bulb of caveolae, and induced caveolar swelling [121,122]. P-selectin, VWF, Angpt2 are good markers of blood ECs [123]. Albondin/gp60 or, a 60-kDa glycoprotein, weakly but specifically binds albumin. Gp60 is expressed in all continuous endothelia, but is absent in fenestrated or sinusoidal endothelia, and in brain capillaries [41,124,125].

## 8. Endothelial Caveolae

One of the important features of vascular ECs is the caveolae (Figure 3A–D). Caveolae can form caveosomes (Figure 3D–F). These structures are not found in enterocytes and other epithelial cells in the intestine, in epithelial cells of the kidney, ureter and urinary bladder, etc. Caveolae were first described in endothelial cells and other cells called micropinocytotic vesicles. Caveolae-like invaginations on the EC PM were discovered in 1953 [126]. In 1955, the term “caveolae,” (from the Latin for “little caves,”) was proposed to describe these invaginations [127].

Caveolae have the appearance of small omega-shaped plasma membrane buds. In the ECs of rat carotid arteries, caveolae are tubule-shaped [128]. Caveolae have a clear spiked coat and a wide caveolar neck open to the extracellular milieu. The coat is positive for caveolin on both the APM and BLPM. This neck appears only after chemical fixation [129]. EM tomography has revealed the protein diaphragm spanning the caveolar opening [129]. Diaphragms composed of the PV1 protein span the caveola opening. The PV1 is N-glycosylated. The coat of the bulb membrane represents a complex of caveolin and cavin family proteins [130,131,132,133]). In the peripheral zone of ECs in tissues, 31.5% of the EC volume (−16, −7.0, and 8.5% of the endothelial cytoplasmic volume) is accounted for by caveolae, their content, and their membranes, respectively. The average density of vesicular openings per μm 2 is 78 in the diaphragm, 89 in the myocardium, 25 in the pancreas, and 10 in jejunal mucosa capillaries [103]. The average neck diameter is 52.2 ± 12 nm. The maximal diameter of a caveola in the section perpendicular to its longitudinal axis is equal to 53 ± 8 nm. The caveolar depth is equal to 48 ± 9 nm [129]. Importantly, after quick freezing of pancreatic islets, caveolae in the ECs of blood capillaries do not exhibit necks [81]. Caveolae are observed within both the central and peripheral zone of ECs. In the central zone of ECs, caveolae are mostly on the APM. Caveolae are rare in blood capillaries in the brain (Figure 3H,I) and in lymphatic vessels (Figure 3F).

In ECs, caveolae and clathrin-coated buds on the PM differ significantly in the following ways: 1. Clathrin-coated pits exhibit rather similar numerical density whereas the density of caveolae in different ECs is extremely variable. They can occupy up to 50% of their PM in the ECs of blood capillaries in the heart and skeletal muscle, or be rather rare (ECs of lymph vessels, or blood capillaries in the brain). 2. Clathrin-coated pits form single pits whereas caveolae form complex higher-order structures (caveosomes) not seen in clathrin-coated pits. The diameter of the caveolae is less than 60 nm. The diameter of PM invaginations coated with clathrin is more than 90 nm. Unlike clathrin-coated buds, caveolae have a protein diaphragm within their neck [130].

In ECs, the caveolae or plasmalemmal vesicles fraction (by comparison with the soluble fraction) have revealed the presence, at high concentrations, of specific caveolar markers, viz., caveolin (both isoforms, with the 24 kDa form being conspicuously more abundant) and Ca^2+^-ATPase. By contrast, angiotensin-converting enzyme and alkaline phosphodiesterase were present almost exclusively in the TX-soluble fraction. The glycoproteins in the vesicle fraction were of apparent molecular weights 52, 68, 95, and 114 kDa. Analysis of the fatty acid composition revealed more palmitoleic and stearic acid in the vesicle fraction then in the TX-soluble fraction. Thus, in comparison with the plasmalemma proper, the vesicle fraction is (1) detergent-insoluble; (2) contains caveolin in two isoforms; (3) contains Ca^2+^-ATPase at high concentrations; (4) contains a set of specific glycoproteins; and (5) is enriched in palmitoleic and stearic acids [134].

The EC caveolae are enriched in GPI-anchored proteins and ganglioside GM1 [135,136,137,138,139], inositol 1,4,5 triphosphate receptor, alkaline phosphatase, and 5+- nucleotidase, a set of specific glycoproteins, annexins II and VI, vimentin, V-ATPase, and the GPI (glycosylphosphatidylinositol)-linked, surface-exposed protein CD59 [138,139,140]. Caveolae isolated from large blood vessel endothelium is enriched in VAMP-2 (and possibly cellubrevin), NSF and SNAP, but not SNAP-23 or SNAP-25 or syntaxin-1 [138,139,140]. Dynamin-2 is not found in all caveolae [141]. Caveolae are highly enriched in cholesterol and sphingolipids. The neck of the caveolae is surrounded by a protein ring that is not made of dynamin [129].

Formation of caveolae in ECs depends on caveolin 1, cavins, availability of cholesterol. Indeed, methyl-β-cyclodextrin induces caveolae disassembly whereas cholesterol-saturated methyl-β-cyclodextrin restores caveolae [128]. Caveolins are oligomeric integral membrane proteins that insert into the cytoplasmic face of the plasma membrane via a putative hairpin domain [128,142]. Caveolin forms a loop, the tip of which is immersed in a lipid bilayer, and both termini are in the cytosol. Due to the concentration of caveolin here, the surface of caveolae has a characteristic relief [129] (Figure 3C).

Caveolin-1 is a membrane protein with both N and C termini in the cytoplasm and stretches of hydrophobic amino acids embedded within the lipid bilayer. In mammals, there are three caveolin isoforms [143,144]. Caveolin-1 is crucial for forming caveolae in tissues other than striated muscle, while caveolin-3 has a similar importance in muscle [145]. In cultured ECs, caveolin-1 is predominantly localized in the GC. Smaller amounts of it are observed within the PM [146].

## 9. Effects Induced by the Absence of Caveolin-1

In total absence of caveolin-1, cultured mouse lung endothelial cells take up albumin (65% of normal rate) mostly through clathrin-coated pits or the clathrin-independent pathway [147]. Gold-conjugated albumin was found to be concentrated within caveolae but not endocytosed by Cav-1-deficient lung endothelial cells. In contrast, aortic segments from wild-type mice showed robust uptake that was time- and temperature-dependent and competed by unlabeled albumin [148,149].

In freshly isolated mouse lung endothelial cells (MLEC) from wild type and cav-1(−/−) mice, ~65% of albumin uptake, as determined by confocal imaging or live cell total internal reflectance fluorescence microscopy, persisted in total absence of cav-1. Uptake of colloidal gold-labeled albumin was evaluated by electron microscopy, and it was demonstrated that albumin uptake in MLEC from cav-1(−/−) mice occurred through caveolae-independent pathway(s) including clathrin-coated pits that resulted in endosomal accumulation of albumin [147].

In caveolin-1-null mice perfused with 5 nm gold-conjugated albumin, gold-conjugated albumin is not endocytosed by Cav-1-deficient lung endothelial cells and remains in the blood vessel lumen; in contrast, gold-conjugated albumin was concentrated and internalized by lung endothelial cell caveolae in wild-type mice [148].

ECs in caveolin-1 knockout mice were completely devoid of caveolae. The transport of proteins (albumin and IgM) and high molecular weight Ficoll was significantly higher in the caveolin-1 knockout mice compared with control mice. The lack of caveolae in Cav-1-KO mice resulted in a marked increase in macromolecular transport. In contrast to wild-type mice, in mice lacking caveolin-1, transcytosis of the albumin was not observed after injection of with gold-labeled albumin into blood [106,148,149]. In ECs lacking caveolin, the TJ morphology displayed no signs of abnormalities. Additionally, tissue cooling did not reduce albumin transport more than could be predicted by the temperature-induced increases in viscosity following cooling. This can be interpreted in four ways: (1) A newly formed, alternative paracellular pathway is opened in Cav- 1-KO mice; (2) the number of large pores in a pre-existing two-pore system is increased; (3) the large pore volume flow in this system is increased; or (4) the selectivity of the endothelial glycocalyx to solute transport is reduced. However, the simplest interpretation is that the caveolae and so called micropinocytosis vesicles derived from caveolae are not transport carriers [150,151].

## 10. Role of Cavins

Cavins polymerize peripherally on top of an array of caveolin oligomers into a network [132,141]. The caveolar neck recruits EHD2 protein [133,152,153,154]. This supports stability of the plasma membranes against rupture [155,156]. In addition, dynamin-2 and Pascin2 are temporarily recruited to the caveolar necks [131,133,157,158,159]. Multiple members of the EHD protein family are recruited to the caveolae neck [156]. Caveolin-2 appears to be less essential for forming caveolae. Mice lacking caveolin-1 do not have caveolae but are apparently healthy and developmentally normal [160,161,162].

Cavins are important structural elements of caveolae. There are four cavins: cavin-1 is needed for forming caveolae in all tissues, cavin-2 and cavin-3 have variable abundance across different tissues, and cavin-4 is muscle-specific [163,164]. Not only caveolin-1, the major protein component in caveolae, but cavins are also required for the formation of caveolae and caveolae-mediated LDL uptake and transcytosis across the endothelium. SR-B1 and ALK1 directly bind LDL and facilitate the transport of LDL through the endothelial cells [165].

## 11. Classification of ECs

Usually, the blood endothelium is classified into (1) closed (continuous), non-fenestrated; (2) closed, fenestrated; (3) porous with small pores; (4) porous with large pores; (5) reticulated; and (6) a simple cuboidal epithelium [107] (Figure 6). We propose the modified classification of vascular endothelium as a whole using several features (Figure 6 and Figure 7).

The first classification feature is BM. ECs can be without BM (e.g., ECs in lymphatic capillaries and in visceral ECs of lymphatic nodes), with a fragmented BM (e.g., ECs in spleen sinusoid, ECs in sinusoid of red bone marrow), with a mesh-like BMs (e.g., lining large arteries), or with a continuous BM (all other ECs). ECs with continuous BM can contain pericytes (blood capillaries in the brain) and seemingly EC containing fenestrae in peritubular blood capillaries in the kidney and in villus blood capillaries in the small intestine. In order to confirm the last statement it is necessary to perform 3D EM analysis. The absence of BM could be caused by the constant penetration of leucocytes through the EC layer or the absence of its synthesis by ECs.

The second feature is their contacts. These contacts may be partially open, as in the sinuses of the spleen (where their function is to test the ability of erythrocytes to be deformed), semi-closed, as in lymphatic capillaries (as described in lymphatic capillaries of the trachea [91] and in the small intestinal villi [81]), or closed as in all other ECs. ECs in the spleen are located along the axis of the vessel parallel to each other and are connected only by their lateral processes. As a result, slit-like elongated holes are formed in the sinus wall between endotheliocytes. The luminal surface of endotheliocytes is mostly smooth, although sometimes there are single micro-processes. Oval nuclei of rod-shaped endotheliocytes protrude into the lumen of the sinuses. Outside, the sinuses are covered with an intermittent basement membrane. There is a solid basement membrane in the areas of inter-endothelial contacts. If red blood cells have increased rigidity, they cannot pass through the opening in the endothelium, remain in the pulp, and are destroyed by macrophages. In the red bone marrow, the sinusoid wall is represented by flattened ECs, in which there are pores of various sizes [10].

The third feature is the presence of pores/fenestrae in the peripheral EC zones [5,108,166,167]. ECs could be continuous or contain fenestrae (in kidney peritubular blood capillaries, capillaries of the cortex and medulla of suprarenal glands, capillaries of pituitary, thyroid and parathyroid glands, pineal body, and intestine villus), small pores (in glomerular capillaries of the kidney), or large pores (in sinusoids of the liver and red bone marrow). Continuous ECs are in lymphatic vessels, endocardium, blood arteries, venules, veins, parietal ECs in lymphatic nodes, in the brain, capillaries of lymph nodes, heart, lung, esophagus, thymus, testis, ovaries, dense bone, and connective, adipose, and secreting mammary gland tissue [108,168,169]. The ECs of nervous system vessels do not contain fenestrae, pores, or trans-endothelial channels [41,170]. The numerical density of fenestrae in different types of ECs differs. The frequency of fenestrae is 1.7 times as high in jejunal (26/μm^2^) as in pancreatic capillaries (15/μm^2^). The frequency of fenestrae appears to be higher and the fenestra fractional area larger in renal peritubular capillaries than in the capillaries of the pancreas and of the jejunal mucosa we have examined [103]. Although in kidney glomerular capillaries, the endothelium is porous, its BM contain very small pores. In liver sinusoids, endothelium is discontinuous with large pores. The BM is purely developed [5,108,166,167].

The fourth feature is the abundance of caveolae. ECs can contain a few caveolae in blood capillaries of the brain (but not in arterioles, where caveolae are abundant; Chow et al. [171]) [80,100,171,172,173,174] (see also Figures 15–24, 15–25, 15–26, 15–27 presented by Rhodin [108]), and in the lymphatic thoracis duct [18]. ECs of the pulmonary trunk have many caveolae [175]. Multicaveolar endothelium has been observed in arteries, blood capillaries of the heart (see also Figures 16–36 presented by Rhodin [108]); striated muscles, dermal blood capillaries [176,177] (see also Figure 2A presented by Braverman [178], the uterus (see Figures 16–38a presented by Rhodin [108], Figure 2A presented by Racho El-Akouri et al. [179], and Figure 1 presented by Baum et al. [180]), adipose tissue (see also Figures 2 and 15 presented by Blanchette-Machie and Scow [181]; Figure 1 presented by Ghinea et al. [182]; Figures 3 and 5 presented by Cinti [183]). Caveolae are observed in the ECs of lung arteries and capillaries as well as well as in ECs of peritubular blood capillaries in the kidney (see Figures 16–37 presented by Rhodin [108]).

The numerical density of caveolae in different types of ECs differs. Caveolae-like structures are observed in hypertensive aortas and mesenteric arteries. However, the numeric density of caveolae in SHR is lower than in normal rats [184]. The small number of caveolae in aortic ECs could be observed after perfusion fixation (Figures 16–57 presented by Rhodin [108]). In Figures 1A-right, B-right, C-right presented by Frias-Anaya et al. [185], there is no apical caveola in brain capillary ECs. In Figure 2c,d from Armulik et al. [100], no HRP is visible in the caveolae of ECs in blood capillaries of the brain. Multicaveolar (see below) ECs in the blood capillaries of the uterus can be seen in the following web site: http://histologyguide.com/EM-view/EM-213-capillary/09-photo-1.html accessed on 30 December 2022). ECs of veins can be classified as an intermediate form [108]. Indeed, in ECs from the saphenous vein, caveolae are visible but their numerical density is significantly lower than in arterial ECs (Figure 3A,B presented by Sugaya et al. [186]).

The fifth feature is the presence of thin peripheral zones. ECs can be thick (as in high endothelial venules in lymphatic nodes) or thin (as in all other organs and tissues [166].

Thus, the following types of ECs can be distinguished (Figure 7I):Thin ECs with open contacts and a fragmented BM (e.g., spleen; Figure 1C);Thick ECs with closed contacts and a porous BM (e.g., high endothelial venules in lymphatic nodes; Figure 1A,B);Thin ECs with semi-closed contacts without a BM (e.g., ECs of lymphatic capillaries and visceral ECs in lymphatic nodes; Figure 1H,I);Thin porous ECs with fenestrae and a continuous BM (e.g., kidney, intestine, endocrine organs; Figure 2A,C–E,I);Thin porous ECs with small pores and a continuous BM (e.g., kidney glomeruli; Figure 2B);Thin porous ECs with large pores and a fragmented BM (e.g., sinusoids of red bone marrow and sinusoids of liver; Figure 2F–H);Thin continuous ECs with many caveolae and a mesh-like BM (e.g., arteries; Figure 1D and Figure 8E–G);Thin continuous ECs with a continuous BM and many caveolae (e.g., multicaveolar ECs in capillaries in the heart, striated muscles, the uterus, connective and adipose tissues; Figure 3A,B);Thin continuous ECs with a continuous BM and a few caveolae (e.g., oligocaveolar ECs of magistral lymphatic vessel and brain capillaries; Figures 1G–I, 3H–J and 5F).

## 12. ECs and Age

Of interest, the fenestrated morphology of the sinusoids is lost with age through the process of “pseudocapillarization”, in which the number of large pores in the liver’s EC sinusoids decreases. Additionally, the endothelium thickens and collagen deposits can be found within the Disse space. The porosity of liver sinusoidal ECs is high [187]. The thickness of BM has also been reported to be significantly increased during aging in microvessels of the brain and retina in mice and humans [188,189]. For instance, the thickness of the BM of the ECs in brain capillaries from 24-month-old mice was double that of 6-month-old mice (107 nm versus 56 nm) [188].

## 13. ECs In Vitro

In cell culture, ECs preserve many of their features but not all of them. ECs cultured on collagen gel form a polarized monolayer, produce a basement membrane, and display Weibel–Palade bodies and caveolae. They are positive for Factor VIII-related antigen and connected by tight junctions [190]. When ECs isolated from different tissue containing many caveolae are grown in vitro, they lose most of their caveolae [191] (see also Figure 4B presented by Montano et al. [192]). Indeed, ECs isolated from blood capillaries of the heart (see Figure 2 presented by Janczyk et al. [191]), from veins (see Figure 3 presented by Zhou et al. [192]) or from the corpus luteum (see Figure 3 presented by Janczyk et al. [191]), have low number of caveolae. The glycocalyx observed in ECs in tissue is almost absent in vitro [193]. However, without apical and basolateral polarization, formation of a vessel lumen is not possible [192]. Additionally, as ECs form contacts in vitro, they also start to produce and deposit BM outside but attached to BLPM [192].

## 14. Functions of ECs

Endothelium forms a barrier between the tissue interstitial space and blood and is responsible for the regulation of the substance passage through this barrier. ECs prevent thrombosis. The endothelium produces a number of vasodilator and vasoconstrictor substances that not only regulate vasomotor tone, but also the recruitment and activity of inflammatory cells and the propensity towards thrombosis. The sub-cellular structure of endothelial cells includes caveolae that play an integral part in regulating the activity of endothelial nitric oxide synthase. Endothelial vasomotor function is a convenient way to assess these other functions, and is related to the long-term risk of cardiovascular disease. Endothelial vasoconstrictors, such as endothelin, antagonize endothelium-derived vasodilators and contribute to endothelial dysfunction [194].

The endothelium-dependent relaxations are due to the release of non-prostanoid vasodilator substances. The best characterized endothelium-derived relaxing factor is nitric oxide (NO). The reduced release of NO from ECs favors the occurrence of vasospasm and thrombosis as well as the initiation of atherosclerosis. NO may affect essential structures within the endothelial glycocalyx [195].

Based on physiological experiments, two sets of pores were postulated to function in capillary and other endothelial cells: small and large pores of about 11 and 50 nm diameters, respectively [196]. It was shown that albumin quickly appears in the pericapillary space of the perfusion of vessels [197], whereas TJs are impermeable to it. Three-dimensional reconstruction demonstrated that the vascular tracer terbium seems to move not through inter-endothelial clefts but rather through endothelial caveolae and clusters of fused vesicles [125]. The intercellular junctions of continuous endothelium were never detectably permeable to tracers of about 1.8–2.0 nm in diameter. However, linearly forded proteins could pass through the TJs [124]. Importantly, during transcytosis the endocytosed material can pass through the GC [27]. The transfer of plasma molecules and lipid-containing particles may be mediated via three different pathways: receptor-mediated transcytosis, fluid-phase transcytosis, and para-cellular transfer [106,198]. The para-cellular transport pathway has been shown to be restricted to macromolecules in the range of 3–6 nm in size [105,106]). In brain blood capillaries, caveolae have a minimal role in the EC transcytosis of nanogold particles, whereas clathrin-dependent endocytosis was involved [173,174,199] (Figure 8A,B). When HRP was used as a marker, large vesicles filled with HRP were observed in the ECs of the brain even under normal conditions, although their presence is rare (see Figure 2C presented by Armulik et al. [102]). However, HRP has not been observed in caveola of ECs of brain blood capillaries [102].

Transcytosis of albumin through ECs is sensitive to inhibition of membrane fusion [135,136,137,138,139,200,201,202,203,204]. The transcytosis of cationized albumin is more efficient than that of native albumin [106,205].

In ECs containing fenestrae, pores, and transcellular channels, these structures are considered the main mechanisms of transcytosis [58]. Fenestrae are found in the capillaries of endocrine glands, digestive tract mucosa, and the kidney (peritubular capillaries). Fenestrae represent pores with protein diaphragms, which have a remarkably constant diameter (62–68 nm). Fenestrae form ordered, parallel linear arrays with an average distance between neighboring fenestrae centers of about 130 nm [103,170]. The fenestra pores have an octagonal symmetry with fibers forming the tufts [206,207]. Heparan sulphate proteoglycans are present only on the luminal side of the fenestra diaphragm [208]. To date, the only gene product localized to the fenestra diaphragm is PV1 [209,210,211]. It is a cationic endothelial glycoprotein encoded by the plasmalemma vesicle-associated protein (PLVAP) gene [212]). The PLVAP product, PV1, was shown to be a key structural component of the FD, necessary for fenestrae formation [206,213]. PV1 is a single-span, 60 kDa, basic, type II membrane glycoprotein that forms homodimers in situ and binds heparin at physiological pH [209,210]. A model of PV1 integration in the diaphragms has been proposed [177]. The extracellular domain of PV1 contains a regular spacing of nine cysteines, four consensus N-glycosylation sites near the membrane, a proline-rich region near the C-terminus, and two large coiled-coil domains. PV1 adopts a rod-like shape. As with all membrane proteins of the PM, this protein is transported through the GC and delivered to the APM or BLPM, and forms rings during the creation of fenestrae [170].

Analysis of super-ultrathin (12–15 nm in thickness) serial sections has demonstrated that less than 1% of seemingly free vesicles appeared as real vesicles [214,215,216,217,218,219,220]. However, these real vesicles could be formed on the GC with the help of COPI or clathrin. A vast majority of the apparently “free” plasmalemma vesicles seen in routine electron microscopic sections are actually part of complex invaginations of the endothelial cell membrane (Figure 3E,F). No free plasmalemma vesicles have been observed [215,216,217,218,219,220]. Sometimes, such clusters are apparently split off and several merged bubbles form a larger formation—a pinosome. In the next step, pinosomes fuse with lysosomes containing hydrolytic enzymes.

Vesicular–vacuolar organelles, described by Dvorak et al. [214], are extremely rare structures. Rare trans-endothelial channels are found with the help of serial ultrathin sections and subsequent three-dimensional reconstruction in ECs of venules and tumor-associated capillaries. They appear as beaded tubules or chains of interconnected vesicles of variable size where vesicles/varicosities are divided by protein diaphragms at the connection points between vesicles/vacuoles and at both exits [221,222,223]. However, no trans-endothelial connections have been demonstrated in quickly frozen ECs of capillaries inside pancreatic islets of Langerhans [129].

## 15. Function of Endothelial Caveolae

In other tissue cells, caveolae do not participate in transcytosis. They perform the function of accumulators of cholesterol and serve as the reserve of the membrane surface. They are also involved in signaling [129]. In spite of this, a completely unusual function has been proposed for the endothelial caveolae as a source of micropinocytic vesicles. It was proposed that they separate from the plasma membrane and, crossing the thinned zones of the cytoplasm of endothelial cells, participate in the transfer of protein through ECs in both directions (both the central and peripheral zone of ECs [114]), and also of lipid particles such as low-density lipoproteins (LDL; [224]). This proposal is based on the analysis of the trans-endothelial passage of albumin. However, careful analysis revealed that in Figures presented in this paper (i.e., see Figure 1 presented by Ghinea et al. [182]), albumin conjugated with gold is visible in round profiles with a diameter of more than 65 nm. The minimal diameter is 70 nm, whereas the bud there has a diameter of 85 nm. In capillary endothelium of adipose tissue after 3 min in situ perfusion of Alb-Au, tracer particles occur only occasionally on the PM. All plasmalemmal vesicles are apparently associated with the luminal front. Caveolae in the BLPM were not labelled [182].

Now, the role of caveolae as trans-endothelial transport carriers is questioned. Caveolae are considered as quite stable membrane structures. They almost do not move on the surface of cells [225]. In living cells, caveolae are mostly immobile plasma membrane domains kept in place by the actin cytoskeleton [226,227]. Caveolae are internalized at a very slow rate [228]. A vast majority of the apparently “free” plasmalemmal vesicles seen in routine electron microscopic sections are actually part of complex invaginations of the endothelial cell membrane (Figure 3E,F). Relative cooling insensitivity of the transfer of albumin across the endothelium in rat lungs does not support the contention of transcytosis of proteins across the endothelium. Neither NEM nor filipin inhibit lung microvascular albumin transport, but actually increase lung endothelial permeability [229].

Caveolae cannot fuse with each other because these do not contain SNAP-25/23 and syntaxin-1 [230]. The caveola-dependent transcytosis should be inhibited when membrane fusion is blocked. Some authors state that the transcytosis is inhibited when membrane fusion is blocked [106,198]. NEM and filipin demonstrated that membrane fusion is important for transcytosis of large particles [231]. Other authors claim that neither NEM nor filipin blocking membrane fusion with the PM caused reductions in albumin (or LDL, see below) clearance across the peritoneal capillaries [151]. NEM markedly inhibits the trans-endothelial transport of small proteins and albumin in cell cultures in vitro and also in perfusion experiments in situ. However, neither NEM nor filipin cause reductions in albumin or LDL clearance across the peritoneal capillaries [151].

In rat aortas treated with N-ethylmaleimide, all caveolae and most free vesicles in the EC cytoplasm, except those around the Golgi area, were HRP-positive. The length density of the abluminal caveolae decreased to about 80% relative to the physiological control level, whereas larger invaginations were more frequently observed. In total, 96.17% of the intercellular clefts were HRP-positive [232].

## 16. Caveolae as the Surface Area Reservoir

Palade [233] noticed that the number of caveolae is much higher than the number of large pores as calculated on the basis of physiological experiments. Presence of caveolae increases the surface area of the PM of ECs from blood-fenestrated capillaries of the islets of Langerhans by up to 70% [129]. Caveolae function as a membrane reserve necessary for ECs, and are found in two states: contraction and stretching [6].

The endothelium of arteries is exposed to pulsatile flow patterns at a relatively high speed (~10 dynes/cm^2^). The elasticity of the vessel wall of conduit arteries helps to convert pulsatile flow in large arteries to continuous laminar flow in smaller resistance arteries. Arterioles and capillaries experience the highest degree of shear stress, at ~50 and 40 dynes/cm, respectively. This decreases dramatically at the venules (~15 dynes/cm^2^), with the lowest values of shear stress occurring in the vena cava (~1 dynes/cm^2^) [234]. This differential shear stress alters the morphology of endothelial cells and can orchestrate planar polarity [13]. As a result, caveolae would protect the plasma membrane of ECs from rupture in the course of cell stretching, not only by external forces such as osmotic pressure, but also by cell–cell and cell–substrate interactions during contraction and relaxation [235]. As muscles are stretched, caveolae become less abundant [236]. Similarly, after perfusion fixation, the number of caveolae in aortic ECs decreases in comparison to immersion fixation [6]). (Figure 8E–G).

In the arteries (including brain arterioles), ECs have a large number of caveolae and should be attributed to the multi-caveolar type of vascular endothelium. Caveolae can flatten as a result of increases in membrane tension. Upon stretching of the plasma membrane, caveolae become flat and liberate cavins into the cytosol and caveolins into the surrounding membrane [237,238]. This phenomenon was characterized by electron microscopy in endothelial cells of muscle capillaries subjected to increased transmural pressure [239]. Chronic exposure (for 1 to 3 days) of ECs to laminar shear increased the total number of caveolae by 45–48% above static control [146]. Thus, one of the most obvious function of caveolae is participation in the extension of the PM. Additionally, near-contact microvilli have a similar function (Figure 6E–G).

## 17. Transport of Large Lipid Particles through ECs

The transport of LDL and HDL through continuous endothelium is the main unresolved issue within this field [240]. Lipids such as free fatty acids, triglycerides, cholesterol, etc. transported in lipid particles should be delivered and transferred to cells, such as adipocytes and muscle cells. Of interest, delivery of fatty acids labeled with isotopes to the adipose tissue is very fast. Already 1 min after intra-venular injection of solution with serum proteins, where free fatty acids were labelled with isotopes, the isotope passed through continuous closed endothelium and was detected in mammary gland tissues [168].

According to the current consensus, LDL and HDL pass the intact ECs through transcytosis by processes which involve caveolin-1, the LDL-receptor, ATP-binding cassette transporters A1 and G1, or scavenger receptor BI [241]. It was stated that the passage of LDL through intact vascular endothelium is a vesicular transport rather than an intercellular diffusion process [242]. Transcytosis through ECs is an active process and causes clathrin-dependent endocytosis [124]. Activin-like kinase 1 (ALK1) or scavenger receptor B1 participate in endothelial LDL transcytosis [165].

Although Balazs et al. [74] proposed that caveolae are responsible for the transcytosis of HDLs, nobody has described HDLs in caveolae. Additionally, nobody has demonstrated vesicula–vacuolar organelles filled with HDL or LDLs [243,244]. Vasile et al. [244] did not show LDL inside 60 nm round profiles or buds during the transport of LDL through ECs. All of the vesicles and buds presented in this paper are larger than caveolae.

In the brain, exogenous horse radish peroxidase was localized in the lumina of blood vessels and in some micropinocytotic vesicles within ECs; none was found beyond the vascular endothelium. Micropinocytotic vesicles were few in number and did not appear to transport peroxidase, while tight junctions between endothelial cells were probably responsible for preventing its intercellular passage. The peroxidase enters an invagination in the cell membrane of an endothelial cell (but does not pass through the cleft between neighboring endothelial cells [245]. Gold particles with the diameter of 4 nm are transported through endosomes [173] (Figure 8A,B).

Of interest, horse radish peroxidase was not seen in small coated vesicles at any interval. Counts of small coated vesicles reveal that during peroxidase absorption they first increase in number in the Golgi region and later, in the apical cytoplasm. COPI vesicles and probably COPI-coated buds have been described on Golgi and clathrin-coated vesicles within the APM. The diameter of apical clathrin-coated vesicles is 83 nm and more [246].

Additionally, albumin (its diameter is 3.56 nm) per se cannot pass through TJs [168]. HDLs (6.5–11 nm in diameter) cannot penetrate through TJs [247]. Additionally, LDL and chylomicrons, which are significantly larger than HDL, cannot do this [106].

Important information has been obtained from experiments with lymph. Lymph taken from skin connective tissue and leg muscles, where ECs of blood capillaries do not contain fenestrae and pores, was essentially devoid of LDL. Concentration of apoB is more than ten times lower than in blood serum, whereas concentration of apoA1 is higher. In lymph, the apoA1/apoB ratios are higher than in plasma [247,248,249]. Total cholesterol concentration in lymph HDL is about 30% greater than in the blood HDL. This means that HDL either specifically penetrates into the interstitial space or is formed there [247]. Elimination of HDL occurs through lymph capillaries [250], because lymphatic capillaries actively absorb lipid particles [81,250].

## 18. Caveolae as Reservoirs for Lipids

The endothelium of the blood capillaries of the heart, striated muscles, and uterus contains a lot of caveolae. Since there is no significant pulse wave in these capillaries and the ECs do not stretch, it is logical to assume that caveolae serve a different purpose here. Since cholesterol is synthesized in the brain, but not in other tissues, with the exception of the liver, it is logical to assume that caveolae can act as a reservoir for cholesterol and fatty acids, which are delivered to the EC through the blood. For the brain, which has its own cholesterol, and lymphatic vessels that remove all lipid particles with a diameter of more than 20 nm from the interstitial space [81] and can receive cholesterol from the lymph, this necessity to take cholesterol from the blood is not relevant (Figure 8C,D).

It is important that the abundance of caveolae is maximal in adipocytes, and in ECs of blood capillaries which deliver lipids to the heart, and striated and smooth muscles. On the other hand, caveolae are enriched in cholesterol because caveolin binds cholesterol [251]. Therefore, it is obvious to propose that they could serve as the site for the temporal storage of cholesterol and fatty acids. The role of caveola as a site for the temporal storage of lipids during their passage across ECs is supported by the observation demonstrating that the uptake of fluorescently LDLs is significantly decreased when caveolin-1 is down regulated in ECs [252].

The number of caveolae in the ECs of lymph capillary in the small intestinal villus increases after feeding and decreases with fasting (our unpublished observations). The number of caveolae in the endothelium of lymphatic capillaries is higher after loading the small intestine with lipids [81]. After appearance of chylomicrons in lymph and blood, the numerical density of caveolae in ECs of thoracic duct increases [18,181] (our unpublished observations). Of interest, in obese rats, the number of caveolae in ECs was increased [253]. Scavenger receptor class B, type I is present in caveolae of brain capillary endothelial cells and expressed almost exclusively at the apical membrane. Disruption of caveolae with methyl-ß-cyclodextrin added from APM results in sorting of this receptor to BLPM [74]. This could explain why caveolin-1 is required for the uptake of LDL by ECs [165].

## 19. Hypothesis

The transport of cholesterol and fatty acids through the ECs can be represented as follows: LDL approaches the glycocalyx, where lipases are located, which cleave VLDL, LDL, and chylomicrons, reducing their size. The transfer of lipids from the external leaflet of the APM to the external leaflet of the BLPM is possible using transcytosis through endosomes (Figure 7(IIB)–(IID)). It is possible also using two flip-flop-like jumps (Figure 7(IIE)–(IIG).

We propose that small LDLs can pass through glycocalyx and contact with the external leaflet of the EC APM allowing cholesterol and fatty acids to enter the external leaflet of the endothelial APM. Additionally, some of the fatty acids attach to albumin and can enter the open caveolae. After that, the fatty acids associated with albumin penetrate into the lipid bilayer, where caveolin is located and there is a lot of cholesterol. There they bind and accumulate.

Then, these lipids jump/flip-flop to the cytosolic leaflet of the APM, pass TJs and diffuse along the cytosolic leaflet of BLPM, and when near caveolae perform flip-flop again into the external leaflet of the BLPM. Excess of cholesterol and fatty acids could be kept in caveolae during this process of membrane based transcytosis.

ApoA is synthesized in the liver and in the ileum and appears in the blood. Apo A1 is secreted (moving along the secretory pathway) from enterocytes and hepatocytes as an unfolded molecule. Therefore, it passes through TJs. Similarly, horse radish peroxidase appeared to permeate endothelial TJs [254]. We propose that this protein, after its passage through TJ, can diffuse within interstitial space and contact with the external leaflet of the EC BLPM. Then, it can take cholesterol and fatty acids from the BLPM of ECs, and form HDL. Then, HDL would deliver lipids to the PM of tissue cells. Elimination of mature HDL from interstitial spaces occurs via lymph capillaries. HDL travels to blood via the lymphatic system along with other macromolecules that exceed the radius of TNF-α (3.24 nm; [247]). In blood, HDLs are captured by scavenger receptors of the hepatocyte BLPM and endocytosed. It is processed by hepatocyte mitochondria into bile acids and thrown into the small intestine. Additionally, excess cholesterol in form of HDL could be removed from tissue into the bloodstream through lymphatic drainage [250]. This explains why the concentration of LDL in the lymph is much higher, and the concentration of LDL is much lower than in the blood. Formation of HDL_E_ particles in the vessel wall has been demonstrated [255].

## 20. Conclusions and Perspectives

In vascular ECs, the structure of the compartments situated along the secretory exhibit typical organization. However, in almost all ECs of adult organisms (the exception being the ECs of blood vessels in the uterus during the secretory phase of uterus cycle), their GC lacks CMC and TMC and contains a significant number of COPI-dependent vesicles. These features suggest in favor of the resting state of the GC. It seems that the intracellular transport per se is not very important for the function of normal adult ECs. However, it is necessary to specify the pathways for the delivery or apically directed proteins and lipids which are very important for the non-thrombogenicity of the luminal surface of the vascular bed. Synthesis and secretion of membrane proteins and proteins of extracellular matrix occur mostly during the G2 phage of cell cycle. Due to rather low mitotic activity, and the preservation of more or less the same thickness of BM for ECs, it is not necessary to synthesize and deliver membrane (for the newly formed two ECs) and extracellular matrix proteins for the formation of the BM. In contrast, compartments of the secretory pathways are deeply involved in transcytosis, especially in vessels lined with continuous thin endothelium with close contacts. Transcytosis occurs with the participation of clathrin-dependent and clathrin-independent endocytosis. The role of caveolae in transcytosis through ECs is minimal.

Our analysis of the existing literature revealed a lot of unclear questions within the field of transcytosis through ECs. Although knowledge about the transport of lipoproteins through ECs is extremely important, it has not yet been well-studied. Our data about the secretion of the components of BM during the G2 phase of the cell cycle are based on regeneration of the aortic endothelium in rat aorta. It is necessary to specify in vitro during what phase of the EC cell cycle the synthesis and transport of BM proteins occurs.

Taking into consideration the fact that the main functions of ECs are transcytosis and protection from blood coagulation, it is obvious to suppose that alteration of these functions lead to pathology. Pathology of EC is related to atherosclerosis, and thrombosis with such dangerous a complication as thrombosis of lung arteries. Biomechanical forces on the endothelium, including low shear stress from disturbed blood flow, also activate the endothelium, increasing vasomotor dysfunction and promoting inflammation by upregulating pro-atherogenic genes. Angiogenesis during cancer is based on collective migration of ECs. Low density lipoprotein cholesterol and oxidant stress impair caveolae structure and function and adversely affect endothelial function [194]. Lipids (in particular LDL) and oxidant stress play a significant role in impairing these functions, by reducing the bioavailability of nitric oxide and activating pro-inflammatory signaling pathways such as nuclear factor kappa B. Regeneration of endothelial wounds depends on expression of VEGFA and turbulent blood-flow-induced influence [256].

Increasing damage of ECs during pathology induces additional synthesis and secretion of components of BM and augmentation of the synthesis of Von Willebrand factor. Indeed, the peri-capillary basement membrane was 38.5 and 45.5% thicker (*p* < 0.05) in people suffering from diabetes mellitus type 2 or intermittent claudication—peripheral arterial disease—than in healthy patients and people with arterial hypertension. The proportion of capillaries with disrupted BM between pericytes and EC was higher (*p* < 0.05) in arterial hypertension patients (33.2%) and people with diabetes mellitus type 2 (38.7%) than in the control patients [180]. Breaks in the TJ bands (crests) lead to brain oedema [80]. It is established that at the onset of collective migration of epithelial cells, Golgi stacks are dispersed to create an unpolarized transitional structure, and surprisingly, this dispersal process depends not on microtubules but on the actin cytoskeleton. Golgi–actin interaction involves Arp2/3-driven actin projections emanating from the actin cortex, and a Golgi-localized actin elongation factor [257]. Nevertheless, the different aspects of the role of ECs in pathology deserve additional analysis.

The numerical density of caveolae is too high to correspond to the calculated number of large pores. Our hypothesis about transcytosis of lipids through the central part of ECs explains not only this discrepancy but also a significant number of experimental observations related to the lipid transcytosis through ECs. Even if our hypothesis is wrong, it could be helpful for the organization and designing of experiments. This hypothesis is based on known contradictions, not on the full range of experiments, but it allows us to outline such experiments. Without it, such experiments cannot be developed. It is better to have a wrong hypothesis than to have none. Finally, we would like to fulfill Popper’s [258] falsification criteria and propose the critical experiments.

Our hypotheses are based on contradictions, not on the full range of experiments, but they allow us to outline important critical experiments. Without such a hypothesis, such experiments cannot be designed. We understand that in the absence of direct confirmation explaining defined controversial phenomena the hypothesis might seem speculative. However, we believe that scientists can still afford some speculation. The publication of such a hypothesis is very useful for science, as it creates a new working hypothesis that attempts to explain the contradictions and that can be tested and either rejected or con-firmed. Moreover, experiments that further reject a hypothesis usually lead to new, more correct hypotheses. Lakatos [259] has demonstrated that in the course of experiments that reject the previous new hypothesis, new, more correct hypotheses usually emerge. Our hypothesis might be interesting for other researchers who could conduct the below listed critical experiments. In any case, these hypotheses need further investigation.

There are several obvious experiments. Injection of the ApoA1 protein tagged with His or other tags into the bloodstream, and examining, with the help of cryo-sections, whether this protein penetrates into the interstitial space in the short time of its circulation would be useful. Additionally, it is feasible to compare the number of caveolae in the EC of the lymphatic thoracic duct, aorta, or blood capillaries of the heart after fasting and after the introduction of a large amount of lipids into the intestine and measure the concentration of HDL and VLDL there. Predictions derive from our hypothesis related to the role of caveolae. Our hypothesis suggests that after starvation the number of caveolae would decrease, whereas after overloading of the intestine with lipids, their numerical density would increase. After feeding, the number of caveolae will increase first via the APM, and then via the BLPM. After loading of enterocytes with lipids, the number of caveolae will be higher on the APM, and after hunger on BLPM. Additionally, after starvation, the decrease in the number of caveolae on the BLPM will be less than the decrease on the APM. When loaded with lipids containing cholesterol, the number of caveolae will increase everywhere except in the EC capillaries of the brain. It would be interesting to know what happens if the cell culture of ECs are incubated with high concentration of LDL. Additionally, it could be important to check whether the secretion of the components of BM occurs during the G2 phase of the cell cycle in vitro.

## Figures and Tables

**Figure 4 ijms-24-05791-f004:**
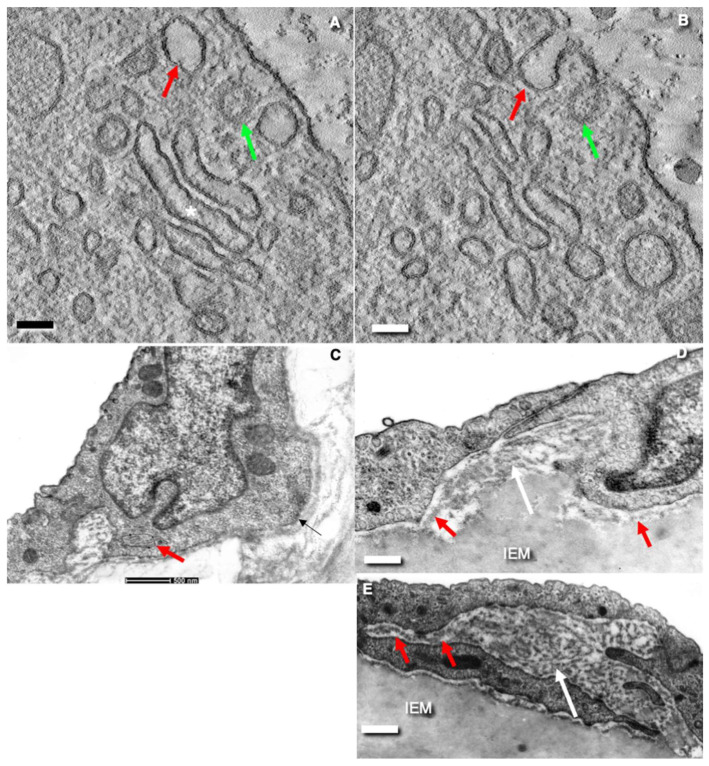
Structure of rat endothelial cells. (**A**–**D**) Serial virtual tomography slices show clathrin-independent invaginations (red arrow in (**A**)) and clathrin-dependent invaginations (green arrows in **B**) of the EC APM. In (**B**) these seemingly isolated vacuoles are connected with the plasma membrane. The Golgi complex (asterisk in (**A**)) is composed of short cisternae. (**C**) EC from rat aorta from the zone of regenerating endothelium. EC is in the G2 phase. Red arrow shows the post-Golgi carrier filled with collagen. (**D**,**E**) ECs of aorta. Red arrows show mesh-like BM. White arrow indicates accumulation of the BM material which is more evident in the ECs from the zone of turbulent flow. Images (**A**,**B**) are taken from Supplementary Figures presented by Sesorova et al. [18]. Images (**C**,**D**) are taken from Figure 5 presented by Sesorova et al. [16]. The image (**E**) is taken from Figure 3C presented by Mironov et al. [6] in agreement with the CC By license. Scale bars: 85 nm (**A**,**B**); 200 nm (**C**); 500 nm (**C**); 305 nm (**D**,**E**).

**Figure 6 ijms-24-05791-f006:**
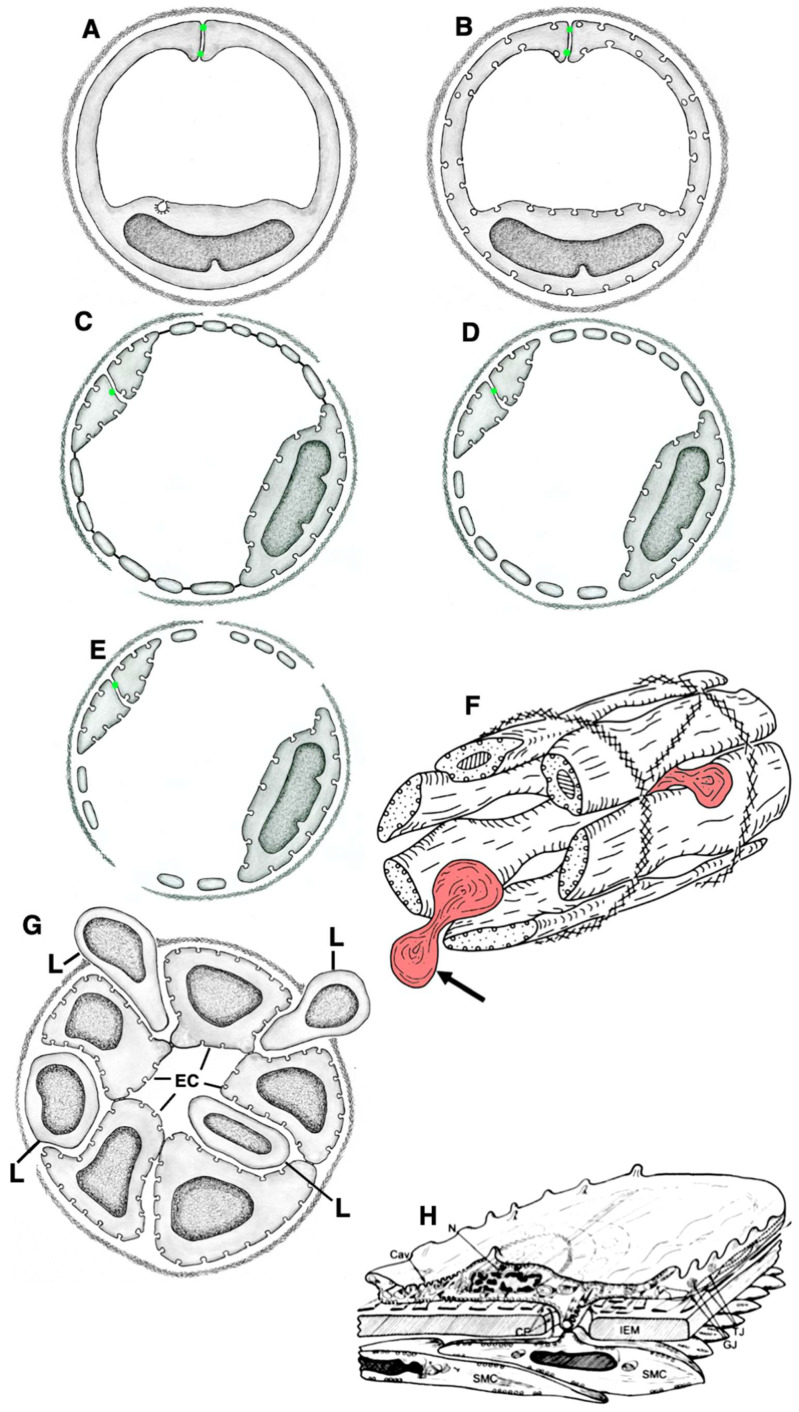
Different types of endothelium. (**A**) The continuous thin oligocaveolar endothelium with close contacts (in brain capillaries and in lymphatic vessels). (**B**) The continuous thin multicaveolar endothelium with close contacts (in blood capillaries of the heart, striated muscles, uterus, and skin). (**C**) The thin endothelium with fenestrae (in renal peritubular capillaries, intestines, and endocrine organs). (**D**) The thin endothelium with small pores (in renal glomeruli). (**E**) Thin endothelium with large pores (in liver sinusoids and red bone marrow). (**F**) Endothelium with open contacts (in spleen). Arrow shows a red blood cell (colored in red) moving through an open contact. (**G**) The cuboidal (high) endothelium (in high endothelial venules of lymph nodes). (**H**) Scheme of the continuous thin endothelium of rat aorta. ECs contain mesh-like BM (below ECs and above IEM). Green dots indicate tight junctions (TJ). Abbreviations: Cav, caveola; CF, collagen fibrils; EC, endothelial cell; GJ, gap junction; IEM, internal elastic membrane; L, lymphocytes; N, nucleus; SMC, smooth muscle cell; TJ, tight junction. Image (**H**) is taken from Figure 2A by Mironov et al. (2020) [6] in agreement with the CC By license.

**Figure 7 ijms-24-05791-f007:**
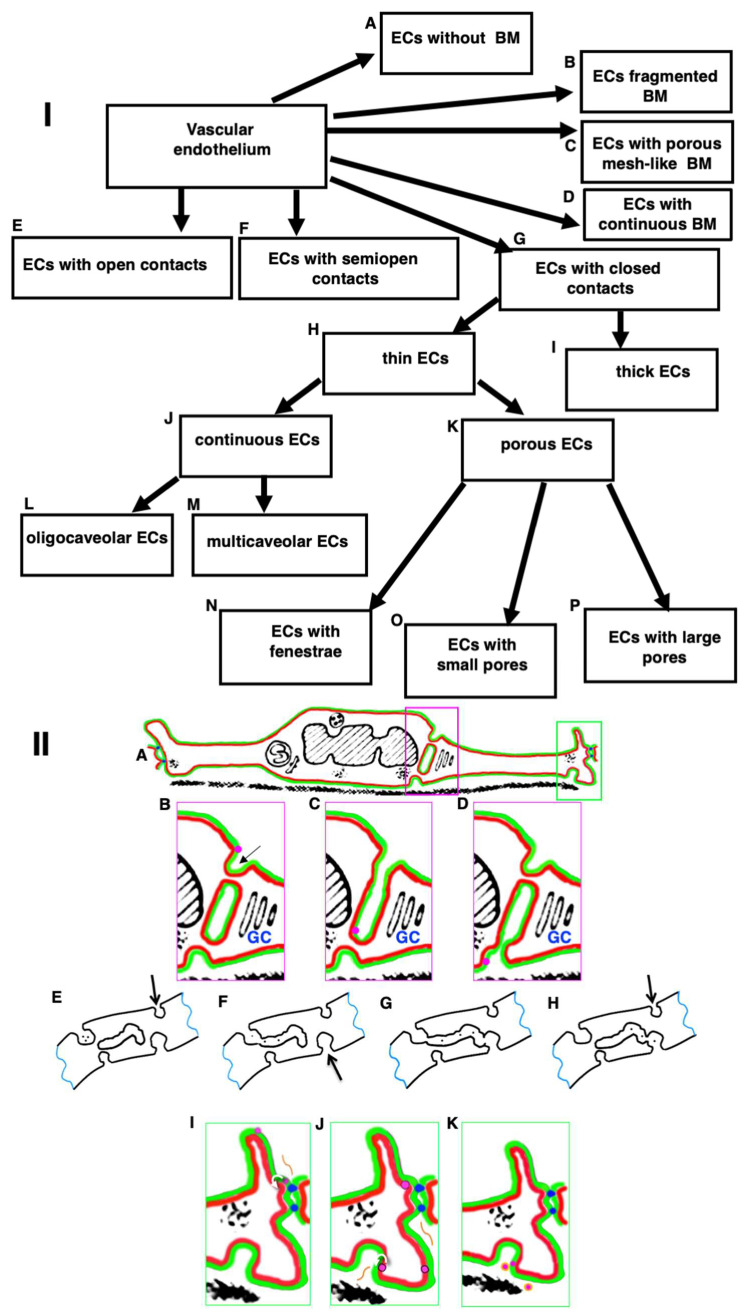
(**I**) Scheme shows principles of the classification of vascular endothelium. (**I**: **A**) ECs without any BM are in some lymphatic capillaries, in capillaries supported cancer and during embryonic development. (**I**: **B**) ECs with the fragmented BM is observed in liver sinusoids. (**I**: **C**) ECs with a porous mesh-like BM are present in large arteries. (**I**: **D**) ECs with a continuous BM are observed in most other vessels. (**I**: **E**) ECs with open contacts are found in the spleen sinusoids. (**I**: **F**) ECs with semi-open contacts are found in lymphatic capillaries. (**I**: **G**) ECs with closed contacts are found in other locations with the exception of those mentioned in (**I**: **E**) and (**I**: **F**). (**I**: **H**) Thin ECs are found in most locations with the exception of those indicated in (**I**: **I**). (**I**: **I**) Thick ECs are observed in high endothelium venules of lymphatic nodes. (**I**: **J**) Continuous ECs can be oligocaveolar (**I**: **L**) in blood capillaries of the brain, in lymphatic vessels) and multicaveolar (**I**: **M**) in arteries, in blood capillaries of the heart, striated muscles, between smooth muscle cells of the uterus). ECs in veins are of intermediate type. (**I**: **K**) Porous ECs can be with fenestra (**I**: **N**), with small (**I**: **O**) or large pores (**I**: **P**). (**I**: **N**) ECs with fenestrae are found in blood capillaries of small intestinal villi, in peritubular blood capillaries of kidney, and in endocrine organs. (**I**: **O**) ECs with small pores are found in capillaries of kidney glomeruli. (**I**: **P**) ECs with large pores are found in the sinusoid of the liver and red bone marrow. (**II**: **A**) Scheme of EC of aorta. The external leaflet of the PM is colored in red. The cytosolic leaflet of the PM is green. (**II**: **B**) If the transcytosis through the central parts of EC functions, then the kiss-and-run mechanism can deliver cholesterol (magenta dots) from the external leaflet of the APM towards the external leaflet of BLPM. (**II**: **C**) Initially endosome fuses with the endocytic bud (arrow) on the APM. Cholesterol diffuses along the cytosolic leaflet of endosomes towards the BLPM. (**II**: **D**) After the endosome fusion with the PM bud on the BLPM and its consequent fission from the APM, cholesterol diffuses towards the BLPM. (**E**–**H**) Scheme of the transcytosis of particles from the lumen towards the interstitial space according to the kiss-and-run mechanism. Both clathrin-dependent and clathrin-independent endocytosis can be involved (see Figure 6A–D). Arrows indicate rare caveolae which are not involved in transcytosis. (**II**: **I**) On the other hand, let us assume that cholesterol (magenta dot) is delivered from LDL to the external leaflet of the EC outgrowth. Cholesterol diffuses towards the TJs (blue dots) between ECs. It cannot pass through TJs. In contrast, a molecule of Apo-A (orange line) can diffuse through TJs. (**II**: **J**) Cholesterol and fatty acids perform flip-flop (curved white arrow) and appear within the cytosolic leaflet of the PM. They diffuse along the cytosolic leaflet of the BLPM towards the bud-like invagination of the BLPM. Simultaneously, Apo-A appears within the interstitial space and diffuses towards the invagination of the BLPM where cholesterol and fatty acids are subjected to flip-flop jump (curved white arrow). (**II**: **K**) Apo-A extracts cholesterol from the external leaflet of the BLPM and forms high density lipoprotein (HDL; magenta dots surrounded with orange border). Then, HDL are extracted from the interstitial space via lymphatic capillaries. Images (**II**: **B**–**D**) represent enlargement of the magenta box shown in (**II**: **A**). Images in (**II**: **I**–**K**) represent the enlargement of the green box shown in (**II**: **A**). The events are shown in dynamics. The image (**II**: **A**) is taken from Figure 2B presented by Mironov et al., [6] and modified in agreement with the CC By license.

**Figure 8 ijms-24-05791-f008:**
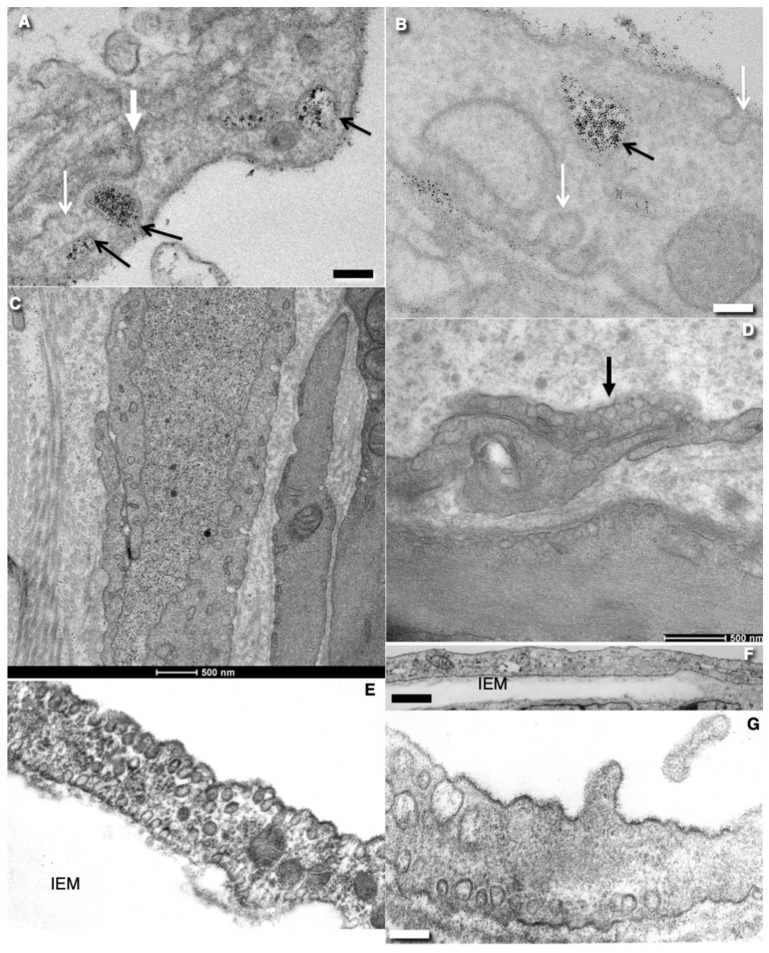
Mechanisms of transcytosis through ECs. (**A**,**B**) The endosome-like structure (black arrow) filled with gold particles. Accumulation of gold particles in the endosomal structure of EC after perfusion of brain with solution containing nanogold particles. Black arrows indicate the endosome. White arrows demonstrate empty caveolae. (**C**) ECs from the lymphatic capillary of small intestinal villus after starvation. Low number of caveolae. (**D**) EC from the same location after feeding with lipids. Accumulation of caveolae (arrow). Images (**A**,**B**) are taken from Figure 4h and Supplementary Figure 3 (the upper right one) (correspondingly) presented by Sanavio et al. [173]. The image (**C**) is taken from Figure 6A presented by Sesorova et al. [18]. The images (**E**,**F**) are taken from Figure 3A presented by Mironov et al. [6] in agreement with the CC By license. The additional images of ECs of blood capillaries within the smooth muscle cell tissue can be found here: (http://histologyguide.com/EM-view/EM-213-capillary/09-photo-1.html accessed on 30 December 2022. The image of pores in the peripheral zone of the EC within the glomerular capillary of the kidney can be found in the following websites: (http://histologyguide.com/EM-view/EM-235-kidney-glomerulus/09-photo-1.html; accessed on 30 December 2022). The luminal surface of a normal aorta can be seen in Figures 2A from Sesorova et al. [16]. The surface of the mitotically dividing ECs can be seen in Figures 4A–G from Sesorova et al. [16]. The structure of the Golgi complex in rat aortic ECs can be seen in Figure 5F,G from Sesorova et al. [16]. The structure of ECs in the G2 phase of the cell cycle can be seen in Figure 5H,I from Sesorova et al. [16]. Scale bars: 110 nm (**A**); 90 nm (**B**); 500 nm (**C**,**D**); 520 nm (**E**); 900 nm (**F**); 430 nm (**G**).

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
