# Peer review of "Intracellular Membrane Transport in Vascular Endothelial Cells"

_ijms, 2023, doi:10.3390/ijms24065791_

Round 1

Reviewer 1 Report

Small typos in line 57 "caveiolae", line 61 "endotheliaum", line 176 "in", lines 289-290 "structures", line 320 "po0lysachharide", line 361 "has" etc. A native English speaker review is definitely recommended.

Reference needed in line 266

Very nice hypotheses of temporary cholesterol-storage site for caveolae.

Conclusions sections could be further developed.

A little more on pathological conditions in which ECs transportation is impaired could be added.

Author Response

Reviewer 1

Comments and Suggestions for Authors

Small typos in line 57 "caveiolae", line 61 "endotheliaum", line 176 "in", lines 289-290 "structures", line 320 "po0lysachharide", line 361 "has" etc. A native English speaker review is definitely recommended.

Reply

We corrected mistakes.

Reviewer 1

Reference needed in line 266

Reply

We ewliminat3d this phrase

Reviewer 1

Very nice hypotheses of temporary cholesterol-storage site for caveolae.

Reply

Thanks a lot.

Reviewer 1

Conclusions sections could be further developed.

Reply

We added future perspectiv3es

Reviewer 1

A little more on pathological conditions in which ECs transportation is impaired could be added.

Reply

We added this

Reviewer 2 Report

In this review, the authors aim to present the data available in the literature concerning the transport of molecules and especially lipids through vascular endothelial cells, especially by the mechanism of transcytosis. Although the topic is interesting the review presents major flaws:

-          Facts are mostly listed without conclusion or discussion

-          Although the text is subdivided into chapters there is no real structure and things are told more than once, mostly without referring to the pervious times.

-          It would be nice to annotate the figure and add more figures to illustrate the mechanisms described in the text. Making illustrations would probably also help to structure the text.

-          Too many abbreviations are used, sometimes without telling what they stand for or telling long after their first use. Thus it’s quite difficult for the reader to understand.

-          There are too many language issues - mistakes, missing punctuation, typos (changing the meaning of the sentence), two words in Russian - for the text to be readable.

Thus, this review should be written again, changing its structure to avoid telling things twice or more. Maybe less facts could be brought together but explained more and coming with a conclusion/discussion from the authors as well as supported by figures – or maybe a table to compare the characteristics of the different type of endothelial cells? - to help understanding. Finally, the new article should be read by a native speaker to avoid mistakes, which make the reading very difficult and sometimes even remove the meaning of some sentences.

Author Response

Reviewer 2

Comments and Suggestions for Authors

In this review, the authors aim to present the data available in the literature concerning the transport of molecules and especially lipids through vascular endothelial cells, especially by the mechanism of transcytosis. Although the topic is interesting the review presents major flaws:

Reviewer 2

-          Facts are mostly listed without conclusion or discussion

Reply

We added discussions to each minichapter.

Reviewer 2

-          Although the text is subdivided into chapters there is no real structure and things are told more than once, mostly without referring to the pervious times.

Reply

We added links

Reviewer 2

-          It would be nice to annotate the figure and add more figures to illustrate the mechanisms described in the text. Making illustrations would probably also help to structure the text.

Reply

We added images and explained them better.

Reviewer 2

-          Too many abbreviations are used, sometimes without telling what they stand for or telling long after their first use. Thus it’s quite difficult for the reader to understand.

Reply

We decreased the number of abbreviations.

Reviewer 2

-          There are too many language issues - mistakes, missing punctuation, typos (changing the meaning of the sentence), two words in Russian - for the text to be readable.

Reply

We corrected mistakes and improved our English.

Reviewer 2

Thus, this review should be written again, changing its structure to avoid telling things twice or more.

Reply

We rewrote the text.

Reviewer 2

Maybe less facts could be brought together but explained more and coming with a conclusion/discussion from the authors as well as supported by figures – or maybe a table to compare the characteristics of the different type of endothelial cells? - to help understanding.

Reply

We added a Table.

Reviewer 2

Finally, the new article should be read by a native speaker to avoid mistakes, which make the reading very difficult and sometimes even remove the meaning of some sentences.

Reply

We corrected our English

Reviewer 3 Report

This is a very well written review manuscript. Covering the problem wide and deep, from histology to the molecular mechanisms that underlie intracellular membrane transport, generaly and particularly in EC. It contains many details, but the whole makes up a valuable article. I found nothing that required correction or inner intervention in the text of the manuscript. In my opinion, this manuscript is suitable for publication in IJMS.

Small things, lines 89 and 93.

Author Response

(The authors gave the same response as above.)

Reviewer 4 Report

The aim of this review is to provide an overview of transport pathways operating in vascular endothelial cells. This is an interesting matter and to my knowledge, only few reviews have been dedicated to this subject. The authors performed an extensive review of the literature and many articles, including old ones, are cited.

Nevertheless, this review can be greatly improved. As it stands, it is quite difficult to read for several reasons.

1) the plan is rather confusing. I would suggest simplifying it. For example, 1) architecture, 2) major transport pathways and their role in establishing polarity 3) transcytosis.

2) it lacks illustrations that would facilitate the reading. For example, a Figure showing the organization of a typical vascular endothelial cell and one showing the main transport pathways would be helpful.

3) The authors write in the abstract that they wish to “re-examine several hypotheses about the role of different mechanisms in intracellular transport..”. One does not understand very well what are these hypotheses by reading the text. The authors could include a Fig. showing which one they favor.

3) it needs English editing. Some sentences are not clear and/or correct. To name a few examples: repeated endothelial regeneration induces appearance the formation, not only…(line 168); the simplest situation with transcytosis through ICs is when ECs have fenestrae (line 268); Most of Golgi stacks are not covered by the cis-most cisterna …; etc…

4) there are (too) many spelling and typo mistakes throughout the text.

Author Response

Reviewer 4

Comments and Suggestions for Authors

The aim of this review is to provide an overview of transport pathways operating in vascular endothelial cells. This is an interesting matter and to my knowledge, only few reviews have been dedicated to this subject. The authors performed an extensive review of the literature and many articles, including old ones, are cited.

Nevertheless, this review can be greatly improved. As it stands, it is quite difficult to read for several reasons.

Reviewer 4

1) the plan is rather confusing. I would suggest simplifying it. For example, 1) architecture, 2) major transport pathways and their role in establishing polarity 3) transcytosis.

Reply

We changed the plan according to recommendation

Reviewer 4

2) it lacks illustrations that would facilitate the reading. For example, a Figure showing the organization of a typical vascular endothelial cell and one showing the main transport pathways would be helpful.

Reply

We added several figures from our previous works.

Reviewer 4

3) The authors write in the abstract that they wish to “re-examine several hypotheses about the role of different mechanisms in intracellular transport..”. One does not understand very well what are these hypotheses by reading the text. The authors could include a Fig. showing which one they favor.

Reply

We re-wrote our abstract

Reviewer 4

3) it needs English editing. Some sentences are not clear and/or correct. To name a few examples: repeated endothelial regeneration induces appearance the formation, not only…(line 168); the simplest situation with transcytosis through ICs is when ECs have fenestrae (line 268); Most of Golgi stacks are not covered by the cis-most cisterna …; etc…

Reply

We corrected these phrases.

Reviewer 4

4) there are (too) many spelling and typo mistakes throughout the text.

Reply

We corrected our English.

Round 2

Reviewer 2 Report

This review endothelial cells got much better. It brings more explanations of the facts and links between them and more illustrations. Especially figure 7 really supports the text about EC classification nicely and helps a lot. Moreover, a personal hypothesis on the lipid transport part is offered as well as a discussion on what could be done next to increase our knowledge about ECs. Due to the aim to bring together all type of data on ECs, the whole paper still seems somewhat unstructured, telling us about ECs, their role, intracellular transport, dysfunction, classification, functions and coming back to transport. However, this may not be a big problem since readers interested in one aspect of EC biology can go directly to the part they want to read. Maybe a table of content would be nice if it’s possible in the review format of this journal. Otherwise, some abbreviations lacking explanation and English language problems remain. I put a few in the word file but may have missed some so please get a native speaker to red the review.

Author Response

We fulfilled all demands proposed by both reviewers. 

Reviewer 4 Report

The manuscript has been significantly improved. However, it requires further editing.

- Summary: put the sentences in the present tense (we propose); this induced necessity..: it is thus necessary to re-evaluate is better; idem for contradictions are accumulated: contradictory results are accumulating.

- Figs. 1-5 need to be carefully checked: the manuscript starts with Fig. 3 which mainly corresponds to paragraph 8!; label ER, Golgi, mitochondria in Fig. 1 G-I.

- There are still too many typos/English mistakes. As a few examples: caveol (line 70); thoracic duct (legend Fig. 1); although they are much lower than MDCK cells (line 151): what does it mean?; there exists (line 220); loose (line 279); could be opened (line 590).

Author Response

(The authors gave the same response as above.)
